

# Studying anomalous propagation over marine areas using an experimental AIS receiver set-up

Laura Rautiainen[1], Milla Johansson[1], Mikko Lensu[1], Jani Tyynelä[1], Jukka-Pekka Jalkanen[1], Ken Stenbäck[1], Harry Lonka[1], and Lauri Laakso[1,2]

[1]Finnish Meteorological Institute, Erik Palmenin Aukio 1, Helsinki, Finland
[2]Atmospheric Chemistry Research Group, Chemical Resource Beneficiation, North-West University, Potchefstroom, South Africa

**Correspondence:** Laura Rautiainen (laura.rautiainen@fmi.fi)

**Abstract.**

Automatic Identification System (AIS) is a wireless communication system used by vessels to exchange real-time information with each other and with coastal authorities, enhancing situational awareness and maritime safety. Consequently, safety at sea depends on reliable signal transmission, which can be disrupted by anomalous signal propagation. In particular, tro-
pospheric ducting can extend the AIS antenna horizon, allowing messages to be received over greater distances than under standard conditions. To study the behaviour of the AIS signal under standard and anomalous propagation conditions, 1-year of AIS-observations were collected from two antennae at 7 m and 30 m heights above the mean sea level on the Utö Island in the Baltic Sea. The AIS antennae were co-located with mast-mounted measurements of temperature and humidity. This allows for studying the AIS signal propagation alongside observed refractivity profiles. The AIS over-the-horizon observations occurred
59% of the time for the 30 m antenna and 34% of the time for the 7 m antenna, mainly during the spring and summer months. A strong diurnal cycle was observed in the Archipelago Sea, north of Utö, while no diurnal cycle was observed in the open sea region south of Utö. During periods of anomalous signal propagation, the AIS messages were received from farther away, from up to 600 km from Utö and the observed received signal strength decayed slower with distance, indicating reductions in propagation losses due to ducting. The anomalous AIS observations were also found to coincide with the stronger and higher
observed ducts.

## 1 Introduction

The maritime situational awareness and safety depend on effective radio communications. The Automatic Identification System (AIS) is a maritime communication system that was developed to prevent maritime collisions and improve vessel tracking (IMO, 2015). It communicates between vessels, and between vessels and vessel traffic services, via 162 MHz channel belonging to the marine VHF range (156-162 MHz). VHF signals propagate via line-of-sight (LoS) propagation and interact
with the troposphere (refraction), and with physical objects and surfaces (reflection, diffraction and scattering). As AIS plays an important role in maritime safety and situational awareness, it is important to assess its functioning under different signal propagation conditions, in particular under tropospheric ducting.





Tropospheric ducting is a phenomenon where due to the vertical gradients of air temperature, humidity and pressure, radio waves get strongly bent towards the Earth's surface, causing them to be trapped in a quasi-horizontal layer known as a duct. Within the duct, the radio waves can travel to further distances than outside of the duct (Patterson, 2008). Due to its impacts on the reliability of signal transmission, ducting has been the subject of study in the field of radio communications for decades (Kerr, 1951). Ducts can be observed in vertical profiles of refractivity which in turn can be derived from vertical profile measurements of air temperature, humidity and pressure from radiosondes (e.g. Ao, 2007; Mentes and Kaymaz, 2007; Basha et al., 2013; Manjula et al., 2016; Liang et al., 2020; Huang et al., 2022a; Qin et al., 2022) and mast-mounted measurements (e.g. Grabner and Kvicera, 2003; Falodun and Ajewole, 2006; Adediji et al., 2011; Wang et al., 2018; Huang et al., 2022b; Rautiainen et al., 2023, 2025). In the near surface applications it is common to use modified refractivity ($M$). Based on the vertical $M$-gradient (dM/dz, $M$ units km$^{-1}$), signal propagation can be divided into four categories: standard refraction (78 < dM/dz < 157), sub-refraction (dM/dz > 157), super-refraction (0 < dM/dz < 78) and ducting (dM/dz < 0) (Turton et al., 1988). The impacts of ducting on the AIS include increased range, interference and signal degradation, and other data anomalies (ITU, 2007). Due to both detrimental and opportunistic effects of ducting (e.g. Norin et al., 2023), developing operative ducting monitoring and forecasting is important to ensure maritime safety and situational awareness.

Besides ducting, other causes of anomalous propagation for the VHF channel include sporadic E propagation, meteor bursts, Earth-Moon-Earth and troposcatter (Rice et al., 2011; Green et al., 2012; Chartier et al., 2022; Sirkova, 2023; Lee et al., 2023). However, at the AIS frequency 162 MHz, troposcatter and ducting are the most relevant factors resulting in anomalous signal propagation. Although troposcatter, i.e. irregularities in the refractive index, can also cause the AIS range to be extended, ducting has been found to cause a significantly greater reduction in propagation losses (Sirkova, 2023). In the northern Baltic Sea region ducting can persist for days (Rautiainen et al., 2025) and is a common phenomenon particularly during spring and summer months (Norin, 2022; Rautiainen et al., 2025).

Due to the scarcity of measurements of vertical refractivity over sea areas, there is a growing need for observations that can be used to validate both waveguide propagation models and ducting forecasts based on numerical weather models. As the propagation of AIS signal is affected by the atmosphere, its propagation characteristics inversely tell us about the properties of the atmosphere it propagated trough. Thus AIS can be a valuable tool for filling the observation gaps over marine areas, particularly as inversion methods improve (Wenlong Tang and Tian, 2019; Han et al., 2022; Huang et al., 2023). While the effects of ducting are well-documented across different radio wave frequencies, its potential impact on AIS signals has garnered limited attention due to AIS being a relatively new system. Prior observation-based research focuses mainly on case studies or point-to-point connections for AIS (e.g. Bruin, 2016; Valčić and Brčić, 2023) and VHF frequency (e.g. Ames et al., 1955; Gunashekar et al., 2006; Constantinides et al., 2022; Chartier et al., 2022). Longer time series are needed to establish the occurrence of anomalous propagation and its diurnal and seasonal cycles. For the northern Baltic Sea region diurnal and seasonal cycles have been previously established based on C-band weather radar ground clutter (Norin, 2022) and X-band surveillance radar over-the-horizon observations (Rautiainen et al., 2025). Particularly if AIS is to be used for validating propagation models, it is important to assess how similar the effects of ducting are across different frequencies and systems. Propagation modelling has been used to study AIS signal propagation under ducting (e.g. Bruin, 2016; Sirkova, 2023).




Applying AIS data for real-time ducting analyses is complicated. The quantity of messages received can be overwhelming, particularly in busy shipping areas. The data itself lacks the information of the transmitting antenna which affects the application of the data unless the information is acquired from the operator of each transmitter, as has been done in Bruin (2016). The data can also include user errors which cause the data to behave in unexpected ways. Furthermore, the reliability of the data can be questioned as received messages can be unintentionally incorrect or intentionally falsified or spoofed (Harati-Mokhtari et al., 2007; Ray et al., 2016). AIS transponder can also be turned off for illicit operations (Mazzarella et al., 2016). Efforts have been put into identifying the false messages in real-time (e.g. Ray et al., 2016; Jaskólski et al., 2021; Wang et al., 2024). Most of the methods involve excluding messages with clearly erroneous data, e.g. invalid Maritime Mobile Service Identity (MMSI), positions over land or unavailable coordinates. More sophisticated methods include e.g. expected-motion prediction and triangularization methods based on the time differences observed between different AIS receivers and multisensor data combining e.g. simultaneous radar and AIS observations.

In this study, an experimental AIS receiver set-up at Utö island in the Baltic Sea is introduced. The set-up includes two receivers installed at different heights (7 and 30 m amsl) that correspond to heights of mast measurements of temperature and humidity (4, 7, 12, 22, 32 and 59 m amsl). This allows for studying the AIS signal propagation alongside observed refractivity profiles. The aim of the study is to (a) introduce the experimental AIS set-up for ducting research and monitoring, (b) explore methods to identify periods of anomalous signal propagation from the AIS data, (c) study how the AIS range and signal strength behave during normal and anomalous conditions, and (d) compare the AIS range with modified refractivity profiles.

## 2 Materials and Methods

### 2.1 Automatic Identification System (AIS)

Automatic Identification System (AIS) is a maritime technology where ships broadcast real-time information to other ships and coastal authorities. AIS was introduced already in the late 1990s but since 2002, the International Maritime Organisation (IMO) has set AIS as mandatory for vessels over 300 GT on international voyages, cargo ships over 500 GT, and all passenger ships, as part of the SOLAS (Safety of Life At Sea) agreement which required these vessels to fit a Class A AIS transceiver (IMO, 2015). Later in 2006, simpler and cheaper Class B AIS transceivers were introduced that can be used on recreational boats. AIS operates by broadcasting real-time data as short binary messages using the 162 MHz VHF frequency reserved for the purpose. The messages do not target specific recipients, but the transceivers process messages from all other transceivers within reach. Terrestrial receivers are set up by authorities or by anyone interested in monitoring the ship traffic. As the range of AIS signal is limited to <100 km under normal propagation conditions, monitoring the marine traffic further away from coast relies on AIS satellites owned by governmental institutions or commercial enterprises.

The AIS messages include vessel's static and dynamic data. The dynamic data is transmitted more frequently than static data. Depending on the ship speed, dynamic data is transmitted from every two (Class A) or five (Class B) seconds when ship speed is over 23 knots to every three minutes at anchor (ITU, 2014). The static data reports are transmitted every six minutes. Currently, there are 27 message types in use. Class A position report includes dynamic data and has message types 1, 2 and



3, while Class B position report has message types 18 and 19. Maritime Mobile Service Identity (MMSI) and International Maritime Organization (IMO) number are used to identify each vessel. A vessel's MMSI comprises of nine digits with the first three digits being the Maritime Identification Digits (MID) (e.g. 2 for Europe and 230 for Finland). For base stations, MMSI

consists of seven digits.

## 2.2    The Experimental AIS Set-up

This study presents the experimental AIS set-up at the Utö Atmospheric and Marine Research Station, located on the island of Utö, in the Archipelago Sea of the Baltic Sea (59° 46'50N, 21° 22'23E). The research station has a long history of atmospheric and marine observations dating back to 1881 (Ahlnäs, 1961; Laurila and Hakola, 1996; Hyvärinen et al., 2011; Laapas and

Venäläinen, 2017; Laakso et al., 2018; Honkanen et al., 2018; Rautiainen et al., 2023, 2025; Honkanen et al., 2024). The region has a lot of potential for AIS based ducting research and monitoring with many busy sea routes near the island and in the study area. The study area is shown in Fig. 1 panel "*a*". The complex archipelago is shown in panel "*c*" based on the European digital elevation model (v. 1.1) from the Copernicus Land Monitoring Service (Copernicus, 2014).

In July 2023, two AIS receiver systems were installed at Utö. Each AIS receiver system consists of VHF and GPS antennae,

receiver, and data logger (Fig. 2). The VHF antennae are Comrod AV7 antennae with vertical polarization and an antenna gain of 2 dBi. The installation heights were set by technical considerations such as location of other close-by radio transmitters and protection against the sea spray. All the antenna cables were set at the same length of 120 m. The mast set-up is shown in Fig. 1 panel "*b*". Later in August 2023, third AIS receiver system was installed at another measurement mast at Utö. However, changes in other instruments on the same mast during the study period caused interference with the receiver and it was excluded

from this study.

Kongsberg AIS RX610 receivers are used for receiving all incoming AIS messages from Channels 1 (161.975 MHz) and 2 (162.025 MHz). The Received Signal Strength Indicator (RSSI) of each message is logged, with the sensitivity of the receiver being –115 dBm which is more sensitive than the typical -109 dBm (ITU, 2007). RSSI is the relative quality of a received signal given in a negative form. The closer RSSI is to zero, the stronger the signal is. The incoming messages are pre-processed

and saved into hourly data files. The messages are decoded in near-real-time using the freely available Python library Pyais for binary AIS message decoding (https://pypi.org/project/pyais/). After decoding, further preprocessing includes discarding messages that have virtual_aid parameter that is non-zero or lack location information. The virtual aid parameter is used to exclude virtual Aid-to-Navigation (AtoN) messages that are simulated by nearby AIS stations rather than broadcasted by the real AtoN (e.g. a buoy). The data used for analysis includes time, message type, MMSI, RSSI and location. Based on the

location, transmitter distance and the azimuth from the receiver are calculated for each message.





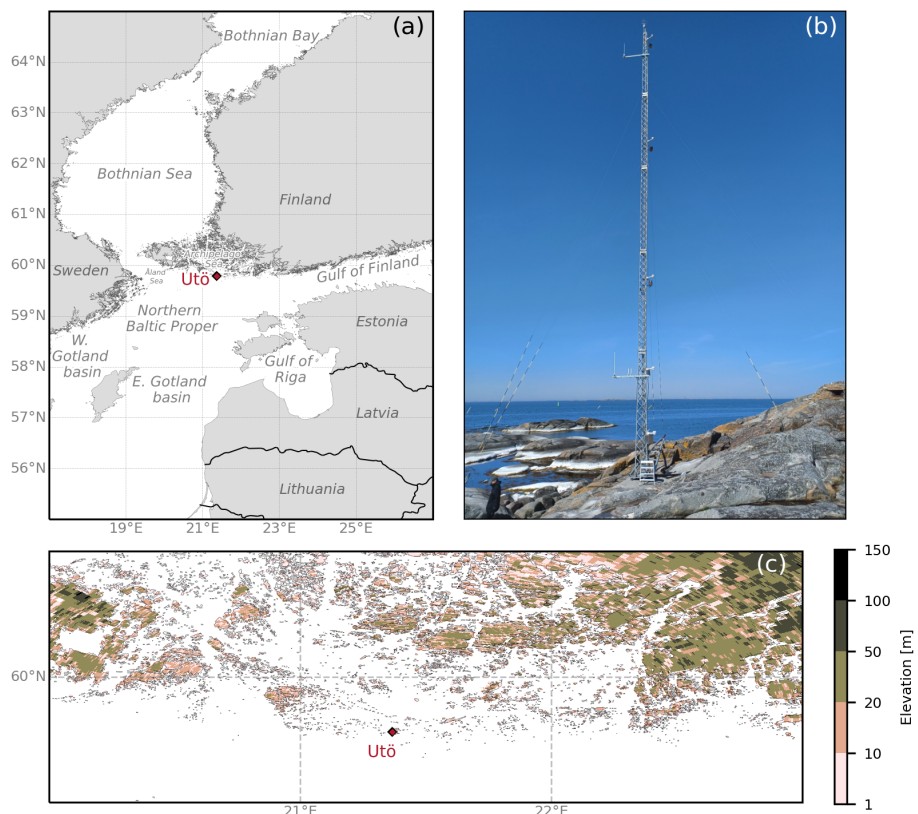

**Figure 1.** (a) The study area encompasses nine Baltic Sea basins. Utö Atmospheric and Marine Research Station is located on the island of Utö in the Archipelago Sea and is denoted by the red diamond. (b) The VHF antennae at 30 m and 7 m amsl on the profiling mast, with two GPS antennae next to the VHF antenna at 7 m amsl. The T-RH sensors at heights 4, 7, 12, 22 and 32 m amsl can also be seen on the mast image. Photo by M. Johansson. (c) Azimuthal presentation (360 degrees) of elevation [m] from Utö given at 100 m spatial resolution.

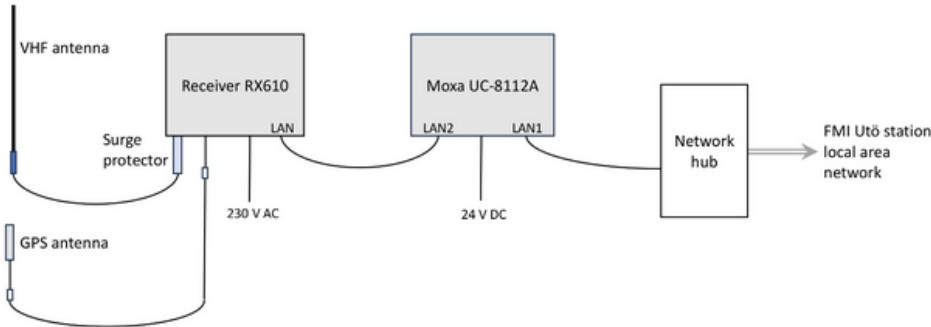

**Figure 2.** An overview of the setup of the AIS receiver and data logger.




## 2.3 Global AIS data

The global AIS data used in this study was provided by ORBCOMM Ltd. The global AIS data is a combination of terrestrial and satellite AIS data. ORBCOMM has a commercial satellite network with 18 AIS-enabled satellites. This results in up to 135 satellite passes and overhead coverage up to 90%. In total 8.9 billion messages were received in 2023. The ORBCOMM global 125 AIS data was used as a background "truth" to establish the areal coverage of the experimental AIS set-up over two months, September-October 2023.

## 2.4 The maximum AIS range

The line-of-sight distance ($R$) is the maximum distance at which a radio signal can be transmitted and received due to the curvature of the Earth under standard refractivity conditions. It depends on the heights of the receiving ($H_R$) and transmitting 130 antennae ($H_T$) and can be calculated:

$$R = \sqrt{2ak} \times (\sqrt{H_R} + \sqrt{H_T}), \tag{1}$$

where $a$ is the earth radius and $k$ is the earth factor that is 4/3 for standard atmosphere (Palmer and Baker, 2006). The line-of-sight distances ($R$) for both 7 m and 30 m antenna are shown in Fig. 3.

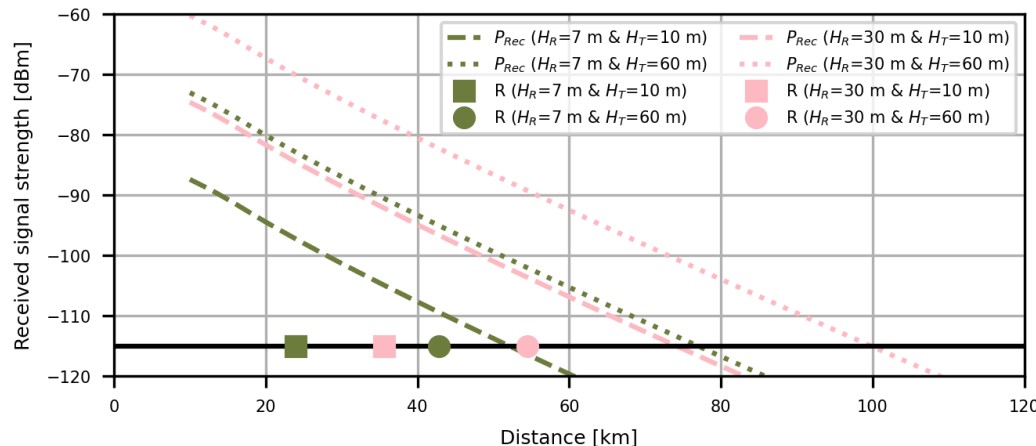

**Figure 3.** Estimations of received power [dBm] for the two receiving antennae at 7 m (green) and 30 m (pink) when the transmitting antennae are at heights 10 m (dashed) and 60 m (dotted). The line-of-sight distances for the two antenna heights are shown for transmitting antenna heights of 10 m (square markers) and 60 m (circle markers). The receiver sensitivity is -115 dBm (solid line).





The role of diffraction is greater at lower frequencies. Hence, the maximum range of terrestrial AIS is governed by line-of-sight and diffraction propagation (ITU, 2007). The smooth earth diffraction propagation loss at 162 MHz can be described:

$$L_{diff} = L_{FS} + F(X) + G(Y_T) + G(Y_R),$$
$$L_{FS} = 20\,log(D) + 20\,log(f) + 20\,log(\frac{4\pi}{c}),$$
$$F(X) = 11 + 10\,log(X) - 17.6 \times X,$$
$$X = 0.029D, \tag{2}$$
$$G(Y_{T,R}) = 20\,log(Y_{T,R} + 0.1 \times Y_{T,R}^3),$$
$$Y_{T,R} = 0.014H_{T,R},$$

where $L_{FS}$ is the free space propagation loss (dB), $f$ is the frequency (Hz), $c$ is speed of light ($ms^{-1}$), $F(X)$ is the distance factor (dB), $D$ is the distance separator, $G(Y_{T,R})$ are the height gain factors (dB) and $H$ is the transmitter and receiver antenna heights (m) (ITU, 2007, 2019a). Received power (dBm) can be estimated:

$$P_{Rec} = EIRP - L_{diff} + G_R + G_{Corr} - L_{misc}, \tag{3}$$

where $EIRP$ is the ship-borne AIS equivalent isotropic radiated power (dBm) typically 41 dBm for Class A vessels, $G_R$ is the receiver antenna gain (dBi), $G_{Corr}$ is the receiver correlation gain (dB) (assumed as 5 dB here) and $L_{misc}$ is the miscellaneous cable losses (assumed to be 1 dB here) (ITU, 2007). The estimated received power (dBm) as a function of distance for the two Utö antennae can be seen in Fig. 3. The maximum AIS range, based on the receiver sensitivity, can be estimated to be 65-80 km for the 7 m antenna and 75-100 km for the 30 m antenna. The range increases with the height of the transmitting antenna.

Information about AIS antenna heights is not readily available and varies between ships as there are no IMO regulations for AIS antenna height or placement on the ship, although it is typically in the mast superstructure above the bridge. For the Baltic ferries this can be up to 50 m. Thus in Utö AIS data, a ship exceeding a certain distance suggested by the range formulas could be either a tall ferry or a small boat observed due to anomalous propagation conditions. AIS data from individual ships with known transceiver specifications could be used to validate the range formulas and study how they can be used to detect anomalous propagation and the associated atmospheric conditions. However, the objective of the present study is to use the AIS data en masse to identify propagation conditions and their seasonal and diurnal variations in the study area extending to nine Baltic basins. Hence, empirical and statistical methods are experimented in order to establish over-the-horizon (OH) propagation criteria and to identify periods of OH observations. The selected basic descriptor for each antenna is the 95th percentile of maximum distance, that is, the radius of a circular boundary such that for 95% of the ships the maximum distance detected during a time period is within the boundary. The 95th percentile was chosen as it was more descriptive of OH observations than the median and less sensitive to individual ships than the 99th percentile. Furthermore, the AIS data was gridded into 0.25° x 0.25° grid and hourly visibility was calculated for each grid and visualised as percentages. This allows for identifying the horizon based on the global AIS product. Hourly visibility is the percentage of hours during which at least one message was received from the grid, and therefore 100% visibility is reached if at least one signal was received every hour during the 1-year period, Aug 2023– Jul 2024.





## 2.5 Modified refractivity and duct characteristics

The various atmospheric properties that influence signal propagation include the vertical stratification of air pressure, temperature and humidity in the atmosphere, which determines the refractivity ($N$) of the atmosphere, i.e. how the signal is refracted and scattered by the air (Turton et al., 1988). Refractivity ($N$) can be estimated:

$$N = \frac{77.6}{T}(P + 4810\frac{e}{T}), \tag{4}$$

where $T$ is the air temperature [K], $P$ is the air pressure [hPa] and $e$ is the water vapour pressure [hPa] (Bean and Dutton, 1968). For practical purposes, modified refractivity $M$ is used near the surface:

$$M = N + \frac{z}{a} \times 10^6, \tag{5}$$

where $z$ is the height above sea level and $a$ is the mean radius of the earth.

In order to study if ducting influences the AIS range observed in Utö, the refractivity $N$ and the modified refractivity $M$ (Eq. 4 and 5) were calculated based on the measurements of relative humidity $RH$, temperature $T$ and pressure $p$ at the Utö Atmospheric and Marine Research Station. Relative humidity $RH$ [%] is related to the water vapour pressure ($e$) by:

$$RH = 100\frac{e}{e_s(T)}, \tag{6}$$

where $e_s$ is the saturation vapour pressure [hPa] (ITU, 2019b). Following an empirical formula, the saturation pressure $e_s$ depends on temperature $T$ [°C]:

$$e_s(T) = a \times exp(\frac{bT}{T+c}), \tag{7}$$

where for the saturation vapour above liquid water $a = 6.1121$ hPa, $b = 17.502$ and $c = 240.97$ °C (Grabner and Kvicera, 2011).

In this study, profiles of the modified refractivity ($M$-profiles) were calculated based on measurements at heights 4, 7, 12, 22, 32 and 59 m amsl. From the $M$-profiles, vertical gradients of $M$ (dM/dh) were calculated. The vertical layer in the $M$-profile where dM/dh<0 is the trapping layer of the duct. The top height of the trapping layer is the duct height. Duct strength (intensity) ($\Delta M$) is defined as the absolute change in $M$ in M-Units (MU) in the trapping layer.

## 3 Results

### 3.1 Number of received messages

The AIS receivers record a large number of messages per day (Fig. 4). On average, after preprocessing, the receiver at 7 m received 198116 valid messages per day and the 30 m receiver 477462 messages per day over the study period. The number of received and preprocessed messages are strongly correlated (r=0.98, p<0.05). During the preprocess, on average 29% of the daily received messages are discarded from the 30 m AIS antenna data and 34% of the 7 m antenna data. The daily percentage



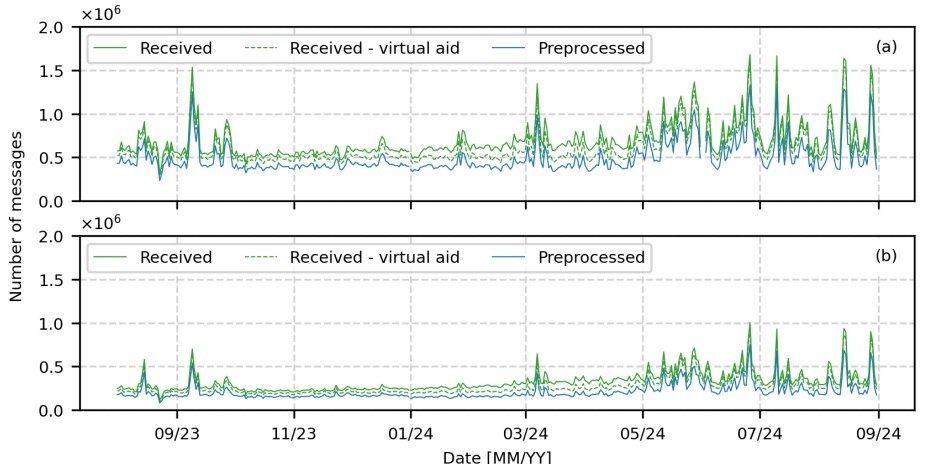

**Figure 4.** Daily number received, received minus virtual aid messages and preprocessed messages for the (a) 30 m AIS antenna and (b) 7 m AIS antenna.

increases during winter and peaks in early spring, nearly at 50%. This is due to the increase in virtual AtoNs over winter due to sea ice in the Baltic Sea (Fig. 4).

The number of messages is relatively stable from October 2023 to February 2024 and unstable from August to September 2023 and March to July 2024 (Fig. 4). The number of valid received messages depends on multiple factors; characteristics of the receiving antenna (e.g. height, sensitivity and power), characteristics of the transmitting antenna (e.g. height and power), any filtering performed in post-processing, the amount of ships within reach, surrounding topography (shadowing) and atmospheric conditions. Majority of these aforementioned factors are stable and any variance in the number of messages is caused by the number of ships within reach, potential shadowing of antennae by other ships and atmospheric propagation conditions. The differences in the number of received messages between the two antennae are most likely due to the lower antenna having a lower range and it being more susceptible to shadowing.

## 3.2 Directional distribution of AIS messages

The directional distribution of Class A and Class B position reports, their RSSI and distance from Utö, in a circular plot (wind rose) divided into eight segments (cardinal and ordinal directions) for two months (Sept and Oct 2023) can be seen in Fig. 5. The percentages correspond to the proportion of the messages that are received from each direction during a month long period. It appears that for October when the daily number of messages was mostly stable, most messages are received from east and northwest, within 0-50 km for the 7 m antenna and 0-100 km for the 30 m antenna. As expected, the share of messages received further away is greater for the 30 m antenna.

Directly to the east is the guest port of Utö, where most of the strongest messages (RSSI > -71) are transmitted from. The guest port is especially strongly represented in the 7 m antenna data, as 30% of the messages are received from within 1 km to



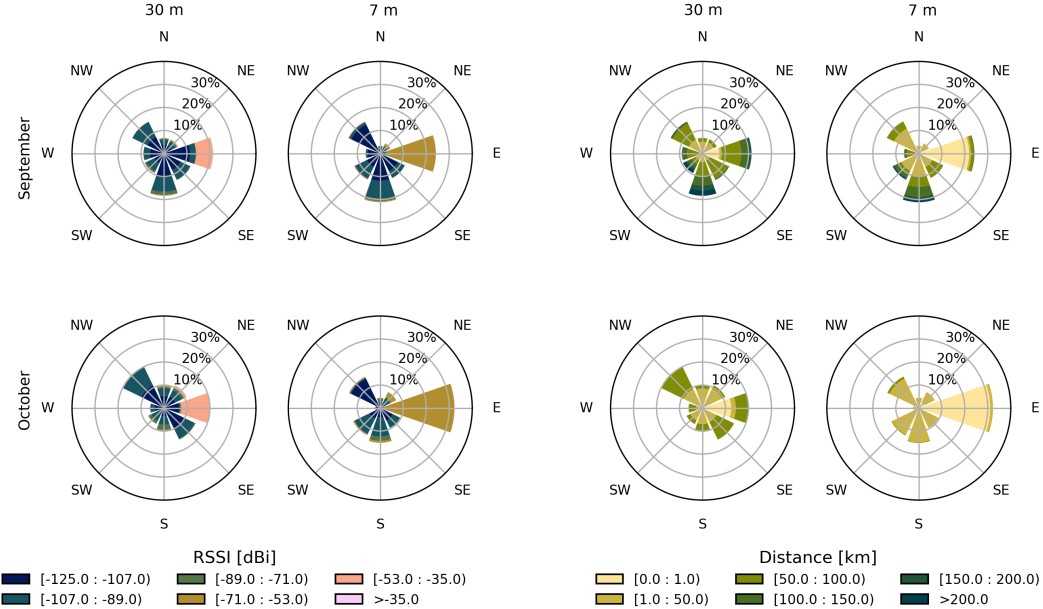

**Figure 5.** Directional distribution of AIS messages and the RSSI [dBi] (two left columns) and distance from Utö [km] (two right columns) of messages received by the two antennae (columns) for two months Sept and Oct 2023 (rows) presented as circular plots divided into eight segments. The percentages correspond to the proportion of the messages that are received from that direction during the month.

the east in October. For September, when the daily number of messages was unstable with some peaks, the share of messages
received from south, and from within the sea sector of the mast (roughly from northwest to southeast) is increased. This decreases the relative share of east in the charts. The share of weaker messages received further away also increases beyond 100 km. This indicates that while the number of messages is increased, some of the messages are also transmitted from further away. The roses for all months during the study period (Aug 2023-Jul 2024) can be seen in Figs. A1 and A2 (Appendix A).

## 3.3 Anomalies in the AIS range

In order to identify anomalies in the AIS range and study their occurrence, the hourly maximum distance of each vessel from Utö was calculated based on Class A and Class B position reports. Based on these maximum distances, daily 5th, 50th (median), and 95th percentiles of distance were then computed (Fig. 6). The nearest 1 km from Utö is disproportionally represented in the data (up to 30%) (Fig. 5) and hence excluded before calculating the maximum distances.

The 95th percentile is more reactive to changes in the AIS range than the median. The time series suggest the presence of a
220 horizon within which the messages are received under standard propagation conditions (within horizon, WH), and periods of anomalous conditions with over-the-horizon (OH) observations. However, the distance to the horizon cannot be expected to be sharply defined as it varies with the characteristics of the shipboard transmitters (e.g. height of the transmitting antenna).



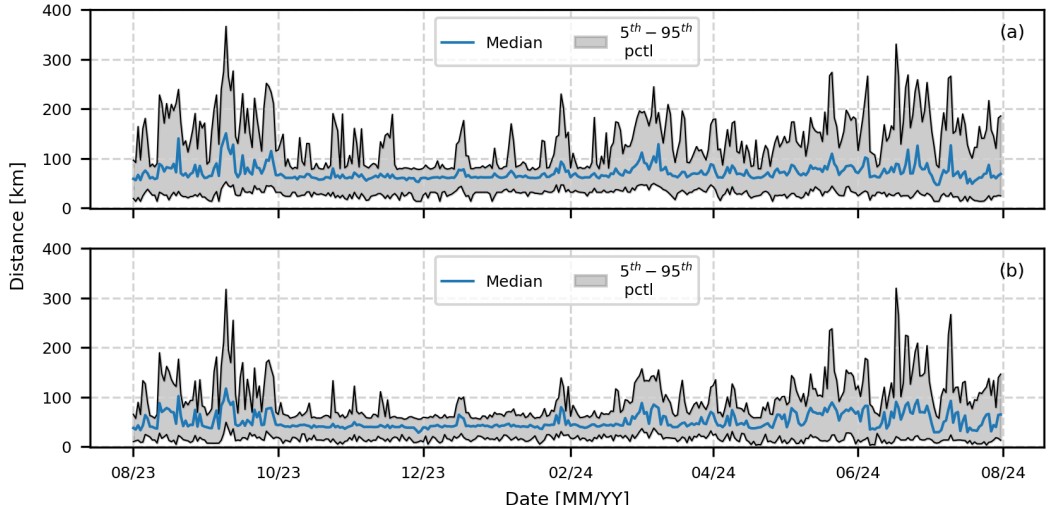

**Figure 6.** Time series of the daily 5th, 50th and 95th percentiles of maximum distance, for the two antennae at the heights of (a) 30 m and (b) 7 m.

Instead, the horizon was studied in terms of a statistical distribution model for the 95th percentile of hourly maximum distance understood as a random variable $\exp(X)$. The data north of Utö was excluded to limit the effects of the archipelago which obstructs signal propagation and restrains the traffic to few fixed routes. This leaves the spatially more scattered traffic in the open sea for the analysis. The model is defined in terms of the logarithm $X$ of the percentile distance. The histogram for $X$ was bimodal for both antenna heights, indicating an overlapping superposition $X = X_1 + X_2$ resulting from standard ($X_1$) and anomalous ($X_2$) propagation. The superposition was resolved by fitting to both components the generalised normal distribution (Subbotin distribution):

$$f(x) = K\exp(-|\frac{x-\mu}{\sqrt{2}\sigma}|^s), \quad K = \frac{s}{2\sqrt{2}\sigma\Gamma(1/s)}. \tag{8}$$

The fitting was done in terms of half-distributions from zero to mode for $X_1$ and from mode to infinity for $X_2$ as described in Rautiainen et al. (2025) where a similar model was used to separate WH and OH components of X-band radar clutter. The result is shown in Fig. 7. The component distributions for the 95th percentile distance, $\exp(X_1)$ and $\exp(X_2)$, are thus of generalised lognormal type (Nadarajah, 2005). The general argument for the applicability of this distribution family is that the decrease of signal power is expected to be a multiplicative result of several factors of different origin.

The superposition weight for $X_2$ is 0.49 and 0.68 for 7 m and 30 m antenna respectively. The parameter $s$ controls the shape of the Subbotin distribution and for $(X_1, X_2)$ has values (1.19,1.68) for the 7 m antenna and (1.08,1.67) for the 30 m antenna. This indicates that especially for $X_2$ the statistical variation is a result of the same process for both antenna heights. The superposition is also apparent in other data types although the bimodality of the statistics is not as clear. For instance, the number of unique ships observed per hour combines the effects traffic density and propagation variations making the separation





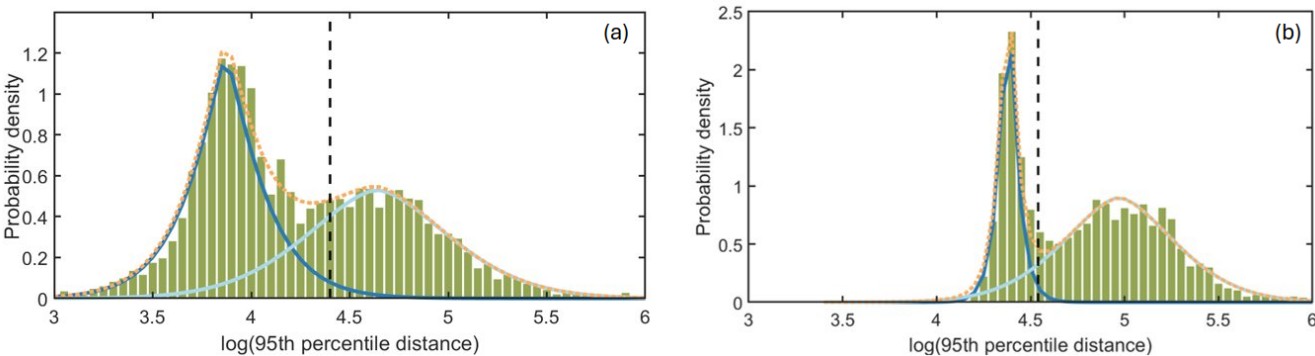

**Figure 7.** The normalized histograms of the 95th percentile distance logarithm for the (a) 7 m antenna and (b) 30 m antenna. The superposition model ($X_1$, $X_2$, $X_1 + X_2$) and horizon (vertical dashed line) are also shown.

of standard and anomalous propagation more complicated. From the number of ships, $\exp(Y)$, the anomalous propagation component $Y_2$ of ship number logarithm for 30 m antenna was separated using quantile-quantile (QQ) plotting for the pair $X_2$ and $Y_2$. The QQ plot was linear for the quantiles corresponding to the half-distribution $X_2$ from mode to infinity. This indicates that the values of parameter $s$ are close to each other and are assumed equal, 1.68. The remaining two parameters of $Y_2$ are

245 obtained from a linear fit to the quantile plot. The results can be seen in Fig. 8. For the separated standard propagation part $Y_1$ the Subbotin fit was less perfect, indicating that the statistics is dominated by traffic density variations. For the 7 m antenna the superposition was barely discernible in the ship number statistics.

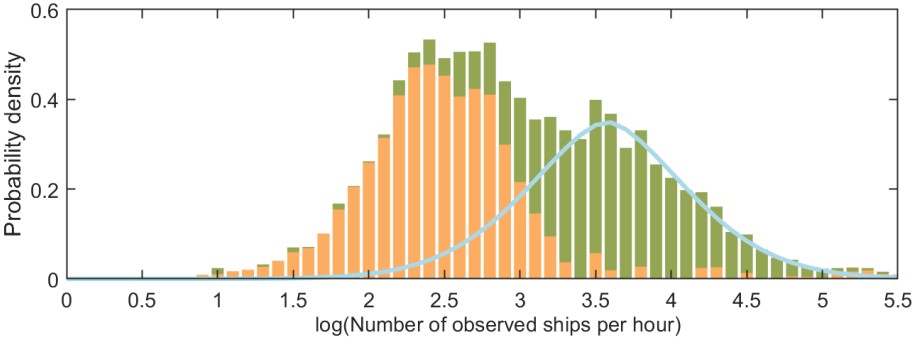

**Figure 8.** The normalized histogram of the number of observed ships per hour logarithm $Y = Y_1 + Y_2$ (green) for the 30 m antenna. The anomalous propagation model $Y_2$ (blue) is shown with standard propagation residual histogram $Y_1$ (orange).

To limit the over-forecasting of anomalous propagation, the statistical definition of the horizon in terms of the distribution model for $X$ was chosen to be the 98th percentile of $X_1$. For the hourly 95th percentile distance this is 82 km for the 7 m

antenna and 94 km for the 30 m antenna (dashed line in Fig. 7). This roughly corresponds to the estimations with a 60 m high transmitting antenna in Fig. 3. It is worth noting that for the 7 m antenna, the 98 percent cumulation of $X_1$ results in a greater





overlap of $X_1$ and $X_2$ (Fig. 7) which means that the chosen horizon will likely under-forecast anomalous propagation when compared to the 30 m antenna.

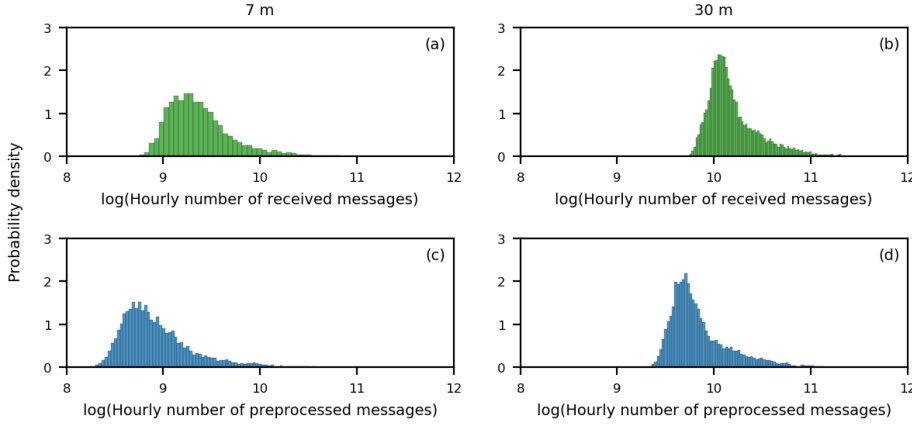

**Figure 9.** The distributions of number of messages logarithm (a) before preprocessing for the 7 m antenna, (b) before preprocessing for the 30 m antenna, (c) after preprocessing for the 7 m antenna and (d) after preprocessing for the 30 m antenna.

For the message number logarithm, the probability distributions are skewed to the right and especially the 30 m case has an atypical tail (Fig. 9). Comparison with Fig. 7 for 95th percentile distances together with the associated analysis, strongly suggests that the distribution shape is a result of a superposition of two symmetric distributions for normal and anomalous conditions, respectively. Unlike for the percentile distance, for which the modes of both component distributions are discernible, the separation of the superposition is not attempted. However, for example the 30 m received message number logarithm clearly indicates anomalous conditions if the value exceeds mode by 0.3 or so. As the message numbers are easily obtained, this can serve as a threshold for commencing other activities during operative monitoring and atmospheric campaigns.

### 3.4 The occurrence of AIS OH observations

For the hourly 95th percentile maximum distance over the 1-year study period, the frequency of OH observations was 59% and 34% for the 30 m and 7 m receiver, respectively, while the corresponding numbers for the data south of Utö were 62% and 33%. From the superposition weights of $X_2$, the selected horizon detects south of Utö 91% and 67% of the anomalous propagation instances for the 30 m and 7 m receivers, respectively. For the 30 m antenna hourly ship number $\exp(Y)$ the values exceeding 37, or the value corresponding to the mode of $Y_2$, are a clear indication of anomalous propagation in similar annual traffic conditions, detecting 50% of the instances. This can be used as a simple indicator of anomalous conditions and occurs 23% of the time.

The seasonal hourly occurrence of OH observations based on the hourly 95th percentile distance is shown in Fig. 10. The OH observations occur more frequently in spring and summer, where the hourly occurrence is up to 90% for the 30 m antenna and up to 65% for the 7 m antenna. During autumn and winter, the occurrence is lower, around 25–50% for the 30 m antenna





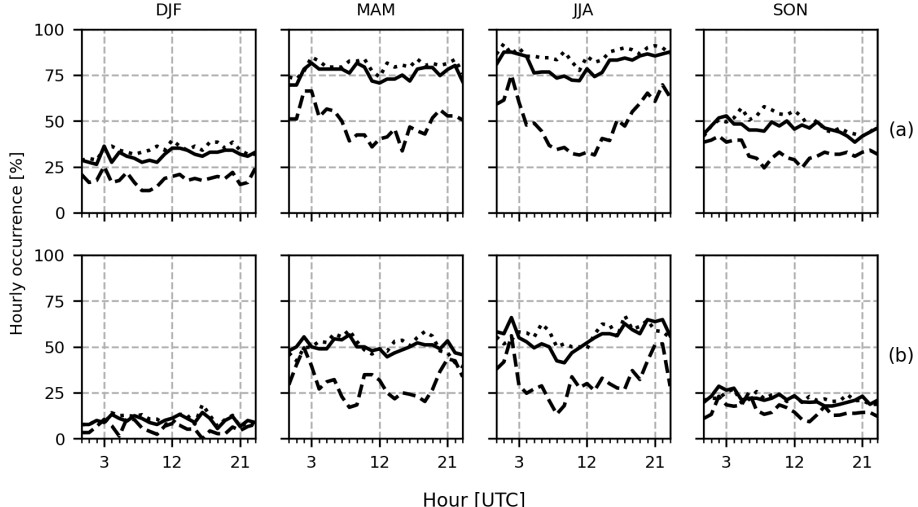

**Figure 10.** Hourly occurrence [%] of OH observations by season for (a) 30 m antenna and (b) 7 m antenna. Solid line is based on all data, dashed line based on the data north of Utö (archipelago), and dotted line based on the data south of Utö (open sea). DJF = December, January and February corresponding to winter, MAM = March, April and May corresponding to spring, JJA = June, July and August corresponding to summer and SON = September, October and November corresponding to autumn

and 0–25% for the 7 m antenna. A diurnal cycle where the OH observations occur more frequently in the evening and at night, and less frequently during the day, is also visible during summer. When the data is separated to the archipelago (north of Utö) and open sea (south of Utö), it is apparent that the diurnal occurrence of ducting is from the archipelago. In the archipelago
sector, ducting occurrence increases 35% from daytime to evening and night. Similar observations were made for the X-band coastal radar in Utö where the strong diurnal cycle was found to result from the archipelago sector where the marine boundary layer is influenced by the boundary layer over land (Rautiainen et al., 2025).

## 3.5    Received Signal Strength Indicator

The daily median RSSI over the study period is around -109 dBi for both antennae (Fig. 11). The hourly median RSSI varies
between -120 and -60 dBi within-horizon, while over-the-horizon, it varies from -120 to -100 dBi. This is explained by the increased reception of weaker messages over-the-horizon. As such, RSSI alone is not a good indicator for AIS OH observations.

    To examine the variability in received signal strength with respect to the distance from the antenna under standard and anomalous conditions, the RSSI over two months, September and October 2023, were plotted against distance, with the number of points illustrated in a 2-dimensional count histogram (Fig. 12). October was chosen because it was identified as the month
that had the least OH observations based on the 95th percentile time series (Fig. 6) and a relatively stable number of daily messages (Fig. 4), while September was identified to have frequent increased OH observations and variable number of daily messages. Only Class A and Class B position reports were included.



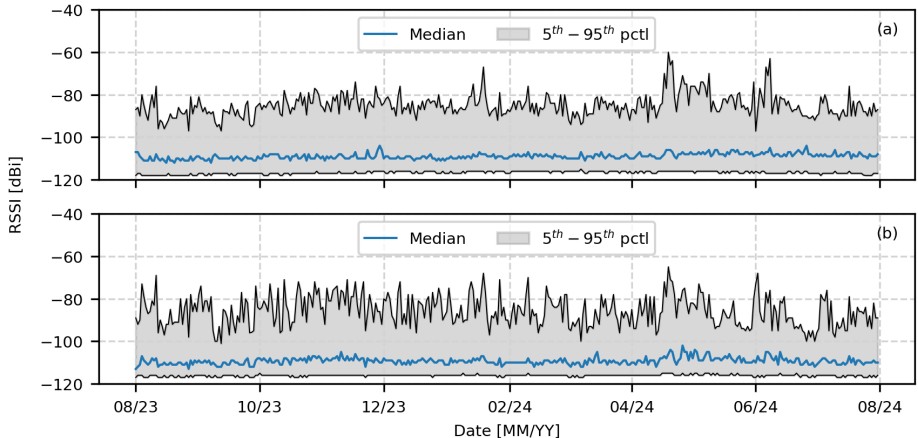

**Figure 11.** Time series of the daily 5th, 50th and 95th percentiles of RSSI [dBi], for the two antennae at the heights of (a) 30 m and (b) 7 m.

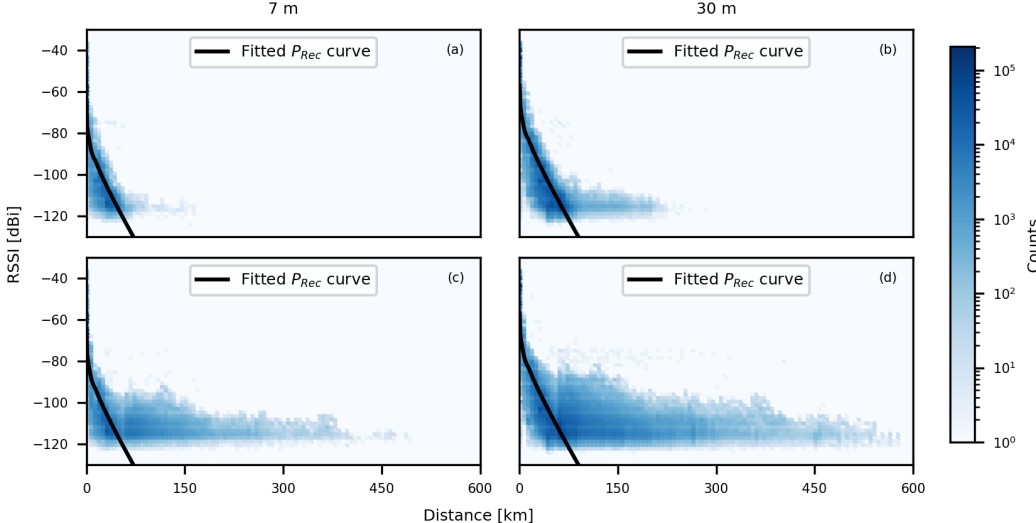

**Figure 12.** Hexagonal binned plot of RSSI [dBi] and the corresponding distance from Utö (km) for the two antennae: 7 m height (left column) and 30 m height (right column). Panels (a) and (b) show data for October 2023. Only message types 1-3 and 18-19 were included. Panels (c) and (d) show the same for September 2023. $P_{Rec}$ curve is fitted to represent the received power under standard conditions over a smooth terrain.

Although October was identified to have the least anomalous signal propagation, it is clear that there are still periods of anomalous signal propagation, although not particularly strong ("a" and "b" panels in Fig. 12). To further assess this, received power ($P_{Rec}$) curve (Eq. 3) was fitted to the October data with time periods of anomalous 95th percentile excluded and the data




limited to the open sea (see Appendix B for more details). The fitted $P_{Rec}$ curve shows the received signal strength indicator (dBi) with distance under standard conditions.

The $P_{Rec}$ curve fits well until the horizon (82 km for 7 m antenna and 94 km for 30 m antenna). When signal is received over-the-horizon, there is no straight-forward relationship between distance and signal strength. This is pronounced when
comparing to September ("c" and "d" panels in Fig. 12) when increased AIS range was frequently observed. It appears that over the horizon, the signal can travel to further distances without degradation, even up to hundreds of kilometers.

## 3.6 Comparison with the global AIS data

The AIS data from Utö antennae was compared with the ORBCOMM global AIS data to establish normal spatial coverage for the antennae. All data were limited to messages received from within the study area (see Fig. 1) and to vessels with MID
starting with numbers 2-7 (regional identifier for individual ships, e.g. first digit 2 stands for Europe). MID was used as the ORBCOMM dataset was very large and provided as separate files for each MMSI.

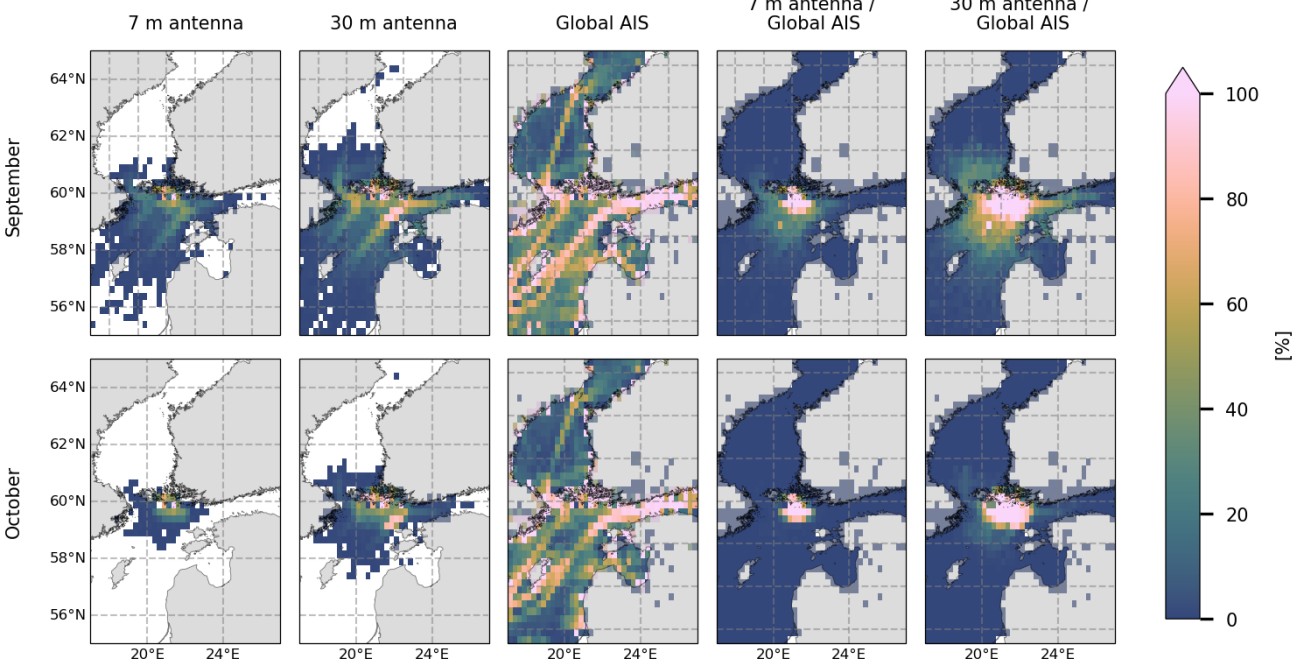

**Figure 13.** $0.25° \times 0.25°$ gridded monthly visibility [%] over two months Sept and Oct 2023 for 7 m antenna (first column from the left), 30 m antenna (second column), global AIS data (middle column) and the ratio between antenna visibility and global data visibility (two right columns). Visibility for a grid is 100% if at least one AIS message is received from within the grid every hour of the month.

The AIS data from the two antennae was gridded into $0.25° \times 0.25°$ grid and the hourly visibility of each grid for September and October 2023 was calculated (Fig. 13). Visibility for a grid is 100% if at least one AIS message is received from within the



grid every hour of the month. The extent of visibility increases with height and both antennae achieve great visibility along the
nearby busy ship routes. The visibility in September is much greater than in October. Although informative, it is unclear if the
regions of low visibility are due to there simply being no vessels to receive messages from, or if the regions are out of range
for the AIS antennae.

To address this, the ORBCOMM global AIS data was also gridded into the 0.25° x 0.25° grid and the visibility of each
grid was presented for the two months (middle column in Fig. 13). The global AIS data was then used as the "background" to
estimate the Utö antenna's coverage area, and the antenna visibility was divided by the global AIS visibility for each grid. The
ratios are shown as percentages (right columns in Fig. 13).

Using the ORBCOMM global AIS as background is beneficial as it shows areas that can be expected to have constant
coverage with the Utö AIS antennae (100% visibility) and regions where coverage is occasionally achieved due to anomalous
propagation (<100% visibility). However, near Utö visibility in some grids exceeded 100%. The global data has a varying
sampling frequency, depending whether the data is collected from the AIS base stations or from satellites. In addition, the
probability of a successful reception of a single sentence message is higher than that of a multi-part message. If $P$ is the
probability, then for multi-part message $P^n$ applies, where $P$ is the probability of receiving a single message and $n$ is the
number of sentences in a multi-part message. For this reason, multi-part message reception probability is lower, especially in
satellite reception of AIS data. As a result, vessels that appear in a grid cell for a short amount of time can occasionally be
missing in the global dataset for that grid. This means that the visibility is likely biased higher and that the bias would likely
increase if the grid size was decreased.

As October had the least amount of ducting, the grids with visibility >95% for October was then used to mask out the grids
within the horizons of the AIS antennae. Each hour of the data was gridded and the number of grids with at least one ship
during that hour was calculated over the study area for each hour (panel $a$ in Figs. 14 and 15).

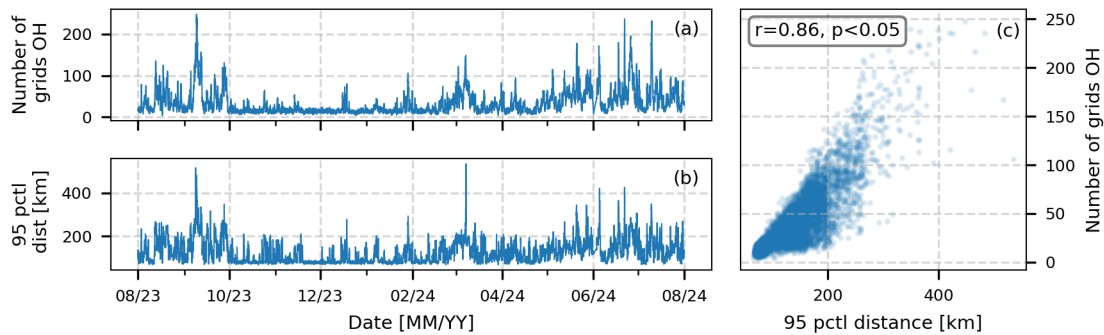

**Figure 14.** Based on the 30 m antenna data: (a) hourly time series of the number of grids with at least one ship outside of the horizon, (b)
hourly 95th percentile of distance and (c) relationship between the two.

The time series of number of grids over the horizon and 95th percentile of distance were compared. The correlation was
high for both antenna heights, r=0.86 for 30 m antenna and r=0.85 for 7 m antenna (panel "$c$" in Figs. 14 and 15). The 95th



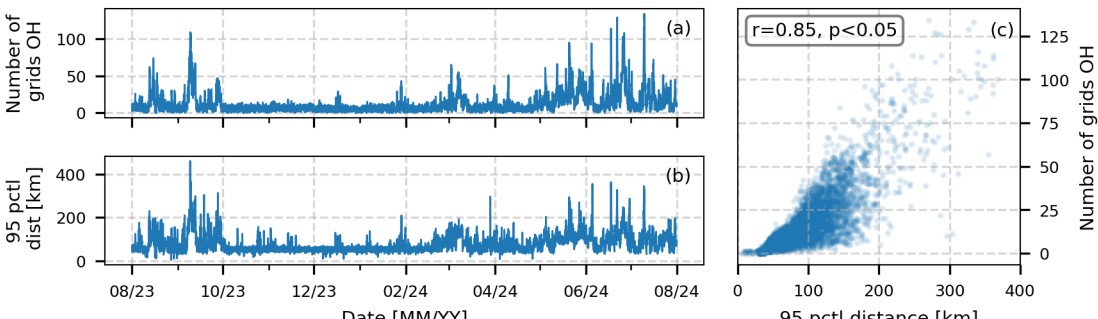

**Figure 15.** Based on the 7 m antenna data: (a) hourly time series of the number of grids with at least one ship outside of the horizon, (b) hourly 95th percentile of distance and (c) relationship between the two.

percentile is more sensitive to the number of ships in the study area and their respective locations, e.g. a small number of ships during an hour could cause a spike in the hourly 95th percentile of distance.

### 3.7 Can the vertical profiles of modified refractivity predict increased AIS range?

In this study we have shown observations of anomalous AIS range. However, it is unclear if the anomalous AIS range is resulting from ducting. Hence, the 95th percentile distance for the two antennae were compared to observed vertical modified refractivity gradients ($M$-gradients) calculated from the temperature and humidity profiles at Utö and to duct strengths calculated from the $M$-gradients (Fig. 16). The time series show that when the 95th percentile distance was at its greatest during summer and autumn, a strong duct was also observed in the vertical $M$-profiles. However, it appears that the 95th percentile distance indicated OH observations more often, particularly over winter and spring, than a duct was observed in the vertical 335 $M$-profiles.

To examine this more closely, the 95th percentile of distance was binned into 10 km interval bins and plotted against how often a duct was observed for each bin (Figure 17). It appears when 95th percentile is at the horizon (90-100 km for the 30 m antenna and 70-80 km for the 7 m antenna), a shallow duct (7-12 m) occurs 20% of the time. When looking at the bins 340 beyond the standard, the share of shallow ducts is smaller and the share of higher ducts (32-59 m) increases, and overall a duct occurs 20-40% of the time. In addition, when the 95th percentile of distance is very low for the 7 m antenna, around 1-10 km, a shallow duct occurs 50% of the time.

Particularly, the highest 95th percentile distances seem to co-occur with the stronger and higher observed ducts. When the duct height is observed at 59 m, the 95th percentile of distance is increased 95% of the time for the 30 m antenna and 90% of 345 the time for the 7 m antenna (Fig. 18). It appears that when the observed duct height is 32 or 59 m, AIS OH observations can be expected.

Although the occurrence of a duct with the height of 32 or 59 m seems to be a good indicator for AIS OH observations, the OH observations still often occur without an observed duct in the vertical $M$-profile. In other words, the measured $M$-profile




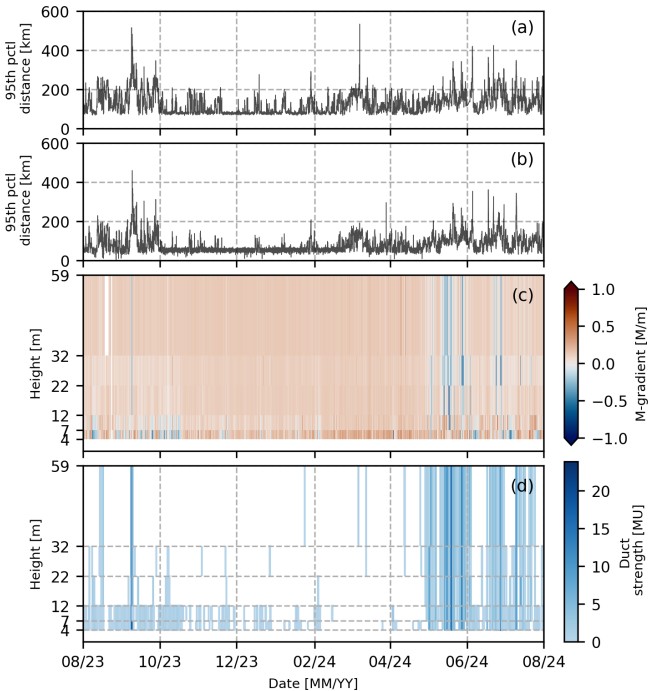

**Figure 16.** (a) Time series of the hourly 95th percentile distance, i.e. the distances outside of which 5% of the received messages originate, for the antenna at the height of 30 m, (b) same as *a* but for the 7 m antenna, (c) vertical modified refractivity gradient over time, for heights 4–7 m, 7–12 m, 12–22 m, 22–32 m and 32–59 m and (d) duct heights (height of a bar), trapping layer thickness (vertical range of the bar) and duct strengths (colour) calculated from the vertical modified refractivity gradients.

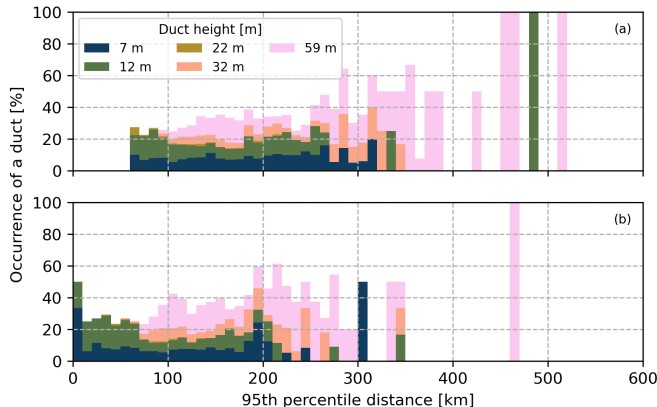

**Figure 17.** The occurrence of duct [%] when 95th percentile distance for (a) 30 m antenna and (b) 7 m antenna are observed at certain intervals. Colors indicate the portions of duct heights.



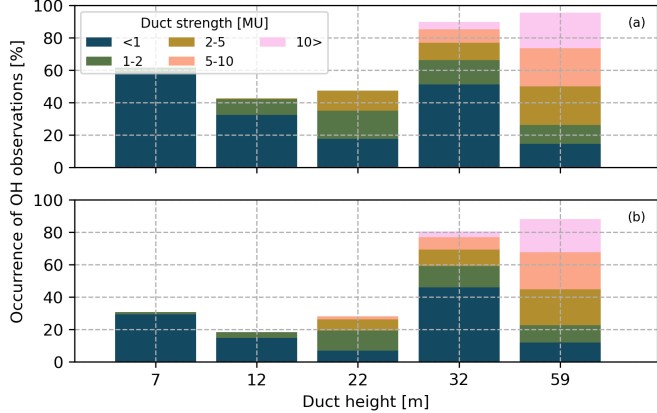

**Figure 18.** The occurrence of AIS OH observations [%] when duct height is at a certain height for (a) 30 m antenna and (b) 7 m antenna. Colors indicate the portions of duct strengths.

under-predicts AIS OH observations. This is likely because the highest measurement height of 59 m might not be high enough

for this purpose, and ducts affecting the AIS signal could have occurred above the highest measurement height. Furthermore, ducts within the AIS horizon, not captured by the measurement mast, could influence the AIS signal propagation.

## 4 Discussion

In this study, an experimental AIS set-up for atmospheric ducting research and monitoring was introduced and approaches for identifying ducting from the AIS data were explored. The approaches ranged from "quick and easy" to more complicated

approaches. First, an approach where simply the hourly number of messages were counted was tested, assuming that ducting would increase the area of reception and therefore the number of messages received. This approach allows to identify peaks in the data without even decoding the received messages. However, the baseline is not stable over time as the number of ships in the Baltic Sea also has diurnal and seasonal cycles, and on longer timescales are affected by the state of economy, e.g. recession (see e.g. Maragkidou et al., 2025). Furthermore, the distribution analysis showed that the distributions are superimposed for the

number of messages and the number of ships and therefore further analysis (e.g. calculating occurrence) based on the number of messages or ships would be complicated and the rate of over- or under-prediction, depending on the chosen threshold, would likely be high.

Secondly, a statistical approach (mean, median and percentiles) based on distance from transmitters to the receiver was tested. The 95th percentile of distance is sensitive to number of messages and hence the data had to be resampled before analysis.

Using the 95th percentile of distance allows for numerically defining the horizon as the WH and OH distributions could be identified and separated. This approach benefits from being comparable to the horizon of the receiver but is not applicable to all directions, particularly to the archipelago sector. In addition, knowledge of the shipping routes within the study area is crucial





when interpreting the results, as the 95th percentile of distance will be sensitive to shipping routes, particularly if the route is curved, causing the distance to receiver remaining the same while the vessel is moving.

Lastly, using a global AIS product as a background truth to establish the horizon was tested. Although this approach takes into account the environment and provides a better result spatially, it is likely that the global data is also influenced by the anomalous propagation conditions, especially when data is collected from the base stations. Further issues might arise from the sampling frequency. Improving the background truth by including SAR-data (see e.g. Li et al., 2022) and fine-tuning grid size, could improve this analysis.

The experimental AIS set-up could provide a cost-effective way to describe ducting conditions over sea areas. While the vertical profiles of refractivity can provide a good estimate of overall ducting conditions, they are not descriptive beyond the local environment, as shown in Rautiainen et al. (2023) where the refractivity profile was found to be descriptive of ducting conditions with the X-band radar over open sea but not in the archipelago. Similarly, in this study the vertical $M$-profiles were found to underestimate the OH AIS observations.

Recently AIS has been studied as a signal source for atmospheric duct parameter inversions (e.g., Han et al., 2022; Huang et al., 2023). These atmospheric duct parameters often include duct height, strength, thickness, and slope. The relationship between AIS signal power and atmospheric duct parameters is complicated and non-linear, thus intelligent optimization algorithms where the atmospheric duct profiles are matched with monitoring data have been utilized (Huang et al., 2023). As these methods keep improving, the potential of AIS for duct monitoring and model validation cost-effectively increases.

Particularly, if the methodology is expanded from experimental set-ups to the operative AIS network, it could, in theory, be used to create a network that defines the signal propagation circumstances for the VHF channel over marine areas in real-time. Based on this network, forecasts of ducting and other signal propagation anomalies could be created. As such, it would be of great value to assess the duct characteristics that influence AIS signal, alongside other frequencies, to assess if the AIS system based forecast could also be applied to other systems, e.g. surveillance and navigational radars, and radio communications.

Comparing the results of this study with previous studies suggests that OH observations occur more frequently for AIS, 34 and 59% of the time, while for an X-band surveillance radar OH observations occurred 19% of the time (Rautiainen et al., 2025) and C-band weather radars' ground clutter in the Baltic Sea region occurred ~10-25% of the time (Norin, 2022).

## 5   Conclusions

In this study, an experimental AIS set-up for atmospheric ducting research and monitoring was introduced. The set up includes
two antennae set at different heights (7 and 30 m) co-located with measurements of air temperature and humidity. This allows for assessing if the near-surface atmospheric stratification affects signal propagation at the AIS frequency.

    A statistical approach, 95th percentile distance, was found to be a good indicator for over-the-horizon (OH) observations with the AIS antennae. The hourly occurrence of OH observations with the antennae was found to be 59% for the 30 m antenna and 34% for the 7 m antenna. The occurrence was more frequent during the spring and summer months. A diurnal cycle where





ducting occurred more frequently during evening and night was found in the Archipelago Sea area (north of Utö) while over the open sea area (south of Utö) ducting was not dependent on the time of the day.

Furthermore, when OH observations occurred, the received signal strength showed less degradation with distance and messages were received from further distances, up to 600 km away. The OH observations were also found to co-occur with the stronger and higher observed ducts. However, the occurrence of ducts underestimated the AIS OH observations.

*Data availability.* Due to the proprietary nature of the data, the collected raw data set cannot be provided on open access bases. The data used in the intermediate level analyses available upon request from the corresponding author.





**Appendix A**

The monthly RSSI and distance of received AIS message types 1-3 (Class A position reports) and 18-19 (Class B position reports) for each compass direction over the study period 08/2023–07/2024.

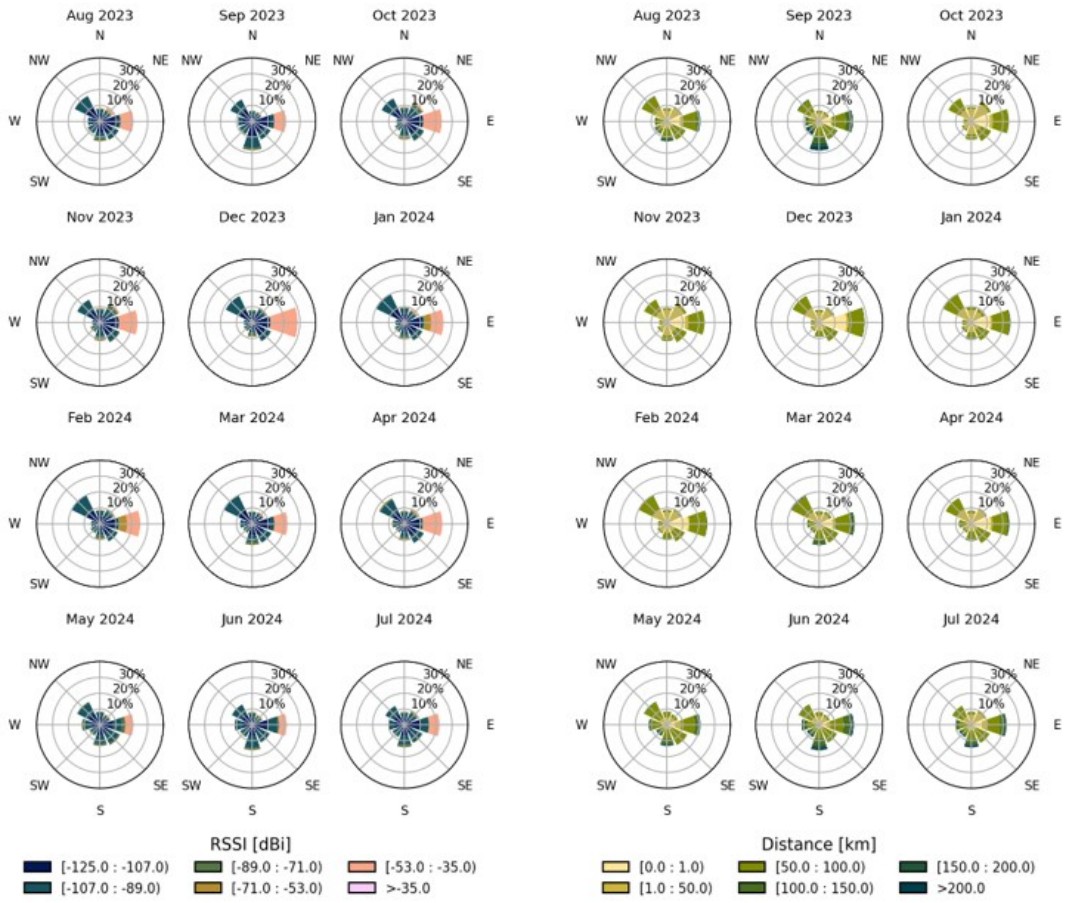

**Figure A1.** Directional presentation of RSSI (three left columns) and distance (three right columns) from Utö (km) of messages received by the 30 m antenna for 12 months.





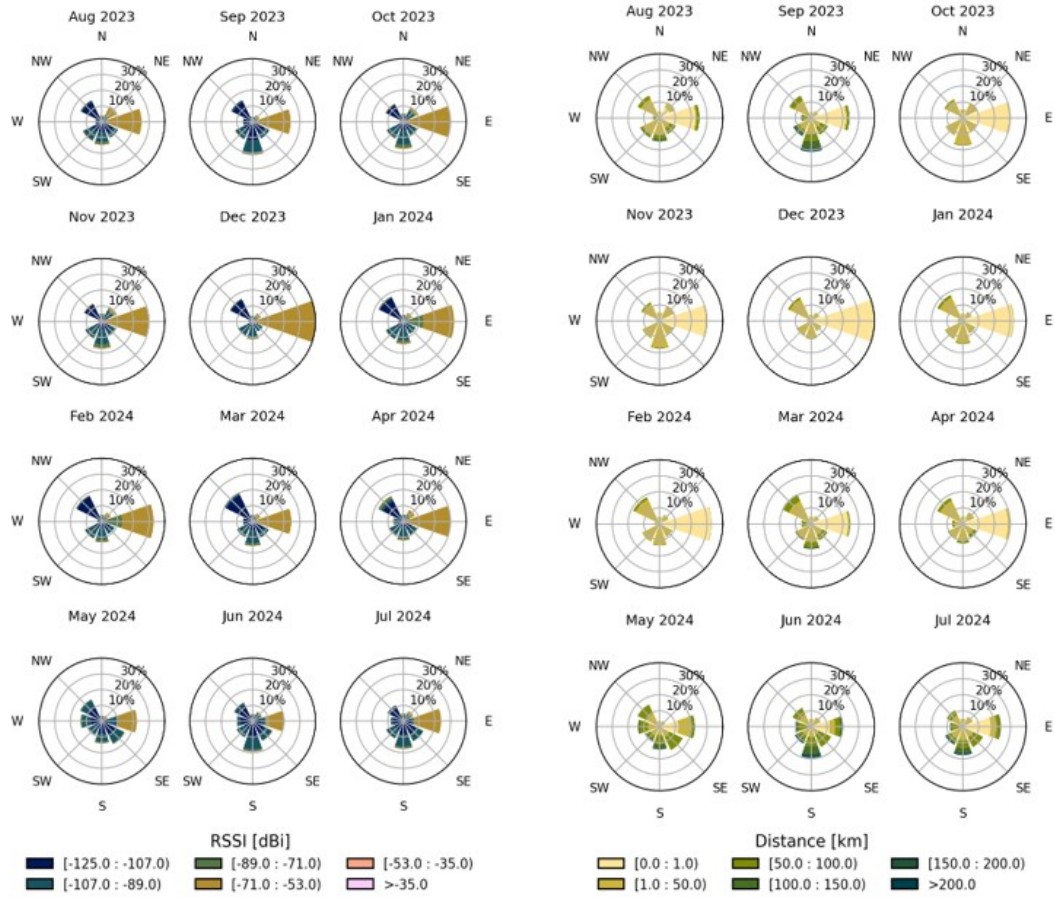

**Figure A2.** Directional presentation of RSSI (three left columns) and distance (three right columns) from Utö (km) of messages received by the 7 m antenna for 12 months.



## Appendix B

In order to establish the received power ($P_{Rec}$) with distance for the two AIS antennae, the received power ($P_{Rec}$) curve was fitted to the October AIS data. To exclude anomalous signal propagation, time periods where the 95th percentile of distance exceeded 94 km for 30 m antenna data and 82 km for 7 m antenna were excluded. In addition, to limit the influence of the archipelago, the data was limited to the open sea region, south of Utö (*a* in Figure B1). As received power is the absolute power in dBm while RSSI is the gain in dBi, a fitting parameter $C$ was added to Eq. 3:

$$P_{Rec} = EIRP - L_{diff} + G_R + G_{Corr} - L_{misc} + C,$$
$$P_{Rec} = -L_{diff} + C.$$

(B1)

The fitted curves can be seen in Figure A1.

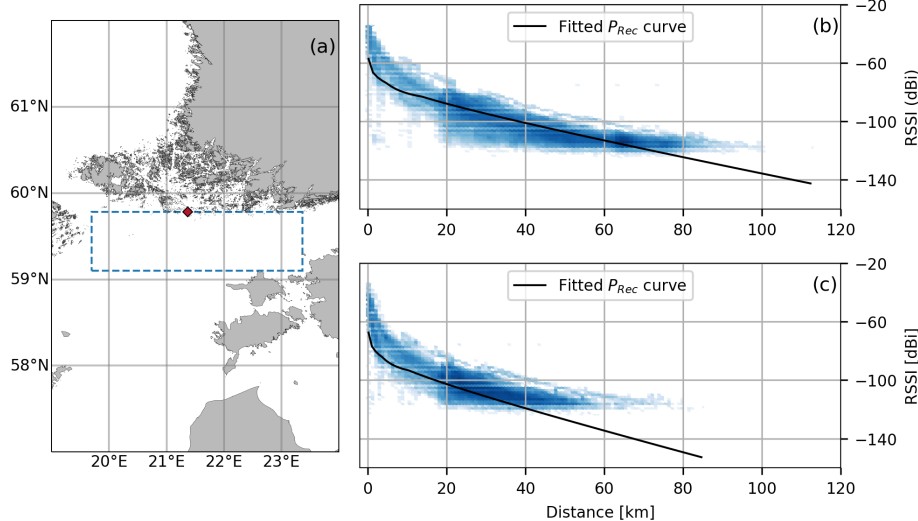

**Figure B1.** (a) The rectangle shows the area that was used for the curve fitting, (b) fitted received power ($P_{Rec}$) curve for the 30 m and (c) 7 m antenna data.



*Author contributions.* All authors contributed to the study through discussions. LL and KS planned and designed the experimental AIS
set-up. KS and HL are responsible for the measurement installations and maintenance. MJ decoded and preprocessed the AIS data and
420 provided the description of the set-up and the diagram showing the set-up. ML performed the fitting of the statistical distribution model
to the probability density distributions, provided the distribution figures and wrote the descriptions. JJ provided the global AIS dataset. LR
performed majority of the data analysis, produced majority of the visualisations and wrote majority of the manuscript with guidance and
feedback from ML, JT, MJ, JJ and LL. All authors provided feedback on the manuscript.

*Competing interests.* The authors declare no competing interests.

*Acknowledgements.* The research was funded by Academy of Finland, project number 338150 "Enabling forecasts on radar performance
in marine environment" and International Cooperative Engagement Program For Polar Research (ICE-PPR). Additional technical or finan-
cial support has been received through Finnish Marine Research Infrastructure (FINMARI), Integrated Carbon Observing System (ICOS),
Academy of Finland, project number 335943 ANTGRAD and H2020 project JERICO-S3 grant agreement No. 871153. This work was
supported by the Strategic Research Council at the Academy of Finland project: Marine waterways as a sustainable source of well-being,
security, and safety (Decision number: 365647). The scientific color maps "batlow", "turku", "bamako" and "vik" by Crameri (2021) were
chosen to avoid exclusion of readers with colour-vision deficiencies (Crameri et al., 2020).



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
