# Peer review of "Studying anomalous propagation over marine areas using an experimental AIS receiver set-up"

_EGUsphere, 2025_

## Author Comment (AC1)

**Response to Referee #1.**

Rautiainen, L., Johansson, M., Lensu, M., Tyynelä, J., Jalkanen, J.-P., Stenbäck, K., Lonka, H., and Laakso, L.: Studying anomalous propagation over marine areas using an experimental AIS receiver set-up, EGUsphere [preprint], https://doi.org/10.5194/egusphere-2025-1790, 2025.

We thank Dr. Alex Chartier for their insightful comments which have helped us to improve and clarify the paper. Below, we have provided a reply to the comments comment-by-comment with the responses in red font.

**RC1 - Alex Chartier - 11 Aug 2025**

Thanks for an interesting manuscript. I have the following comments, but defer to tropospheric experts regarding the significance of the results.

1. What is the distribution of propagation distances observed? Can you show a histogram of distances to illustrate the 'break point' between normal and anomalous propagation? The selected criterion (95th percentile of maximum distance) seems ad-hoc and vulnerable to variations in the distribution of ships with relation to the receiver station (as noted by the authors between lines 325-30). Why not use a simple distance cutoff (e.g. at least X counts >300 km indicates anomalous propagation)?

Besides atmospheric conditions, the propagation distance is affected by the heights of transmitting and receiving stations and the power of the transmitting station. The height variation is very large as the traffic includes ships of all sizes from small tugs to large ferries for which the bridge can be more than 40 meters from the sea level. Also, the anomalous propagation conditions are seldom uniform over any larger area in the Archipelago Sea. This is clearly seen from coastal radar data where distant targets over the normal horizon flicker, disappear out of sight, and reappear during anomalous conditions.

In order to use distances to the transmitting ships unambiguously to quantify propagation, the data should be normalised by the transmitting power and the antenna height, where the latter can be assumed to have more variation. However, the antenna height data is not included in the marine radio station information databases but must be requested from the shipping companies or estimated from particulars data or images (e.g. using bridge roof height). We are aware of the

potential of such data to reveal the spatial distribution of propagation conditions and have plans to utilise it in the future. However, any definition of distance behind which the reception is interpreted as following from anomalous conditions, based on statistical distributions or other considerations, is bound to be ship specific.

We are also aware that the normal and anomalous conditions manifest as a superposition appearance in the distance histograms (See Figure 1 below). However, the distance data is not as suitable for our analysis that is targeting the identification and classification of anomalous conditions, and the percentile data that was chosen precisely to have a descriptor that is less sensitive to the variation of transmission parameters. The use of percentile data also connects our AIS based research with our earlier radar-based work (Rautiainen et al, 2023, 2025). Also for the radar data, the properties of the island and ship targets (reflectivity, height) make the distance-based measures less applicable. We also find the applicability of the same distribution superposition model for both AIS and radar data an argument in favor of our approach.

Here's an example of the distribution of propagation distances observed with the 7 m antenna over September 2023:

Figure 1. Histogram of distance [km] for the 7 m antenna over September 2023. The data is limited to Class A and Class B position reports from within the study area.

The closest 1 km is excluded.

Additionally, the goal was to achieve a metric that limits the amount of data while being descriptive of the visibility. As the antennas receive on average 200 000-500

000 messages per day, limiting the number of data points to 24 per day before doing further analyses was desirable. The above histogram consists of 2.3 million data points alone.

Prior to submitting our paper, we tested the effect on the results by changing the 95th percentile to median, 85th percentile, and 99th percentile. The distributions for the median, 95th percentile, and 99th percentile are very similar. The 99th percentile was more sensitive to individual ships while the median was not very descriptive of the visibility, hence we decided to use the 95th percentile.

We hope this clarifies why the metric was chosen.

Rautiainen, L., Tyynelä, J., Lensu, M., Siiriä, S., Vakkari, V., O'Connor, E., Hämäläinen, K., Lonka, H., Stenbäck, K., Koistinen, J., & Laakso, L. (2023). Utö Observatory for Analysing Atmospheric Ducting Events over Baltic Coastal and Marine Waters. *Remote Sensing*, *15*(12), 2989. https://doi.org/10.3390/rs15122989

Rautiainen, L., M. Lensu, V. Vakkari, J. Tyynelä, H. Kanarik, and L. Laakso, 2025: Marine Atmospheric Ducting Statistics Based on 2 Years of Coastal Surveillance Radar Observations. *J. Appl. Meteor. Climatol.*, **64**, 63–76, https://doi.org/10.1175/JAMC-D-24-0096.1.

It is not obvious (at least to me) whether the results are in keeping with what is
expected from current atmospheric propagation models. Additionally, the ducting
analysis is restricted to local conditions at the receiver site (Utö). These two issues
could be remedied by comparing the results to duct strengths calculated from
meteorological reanalysis data.

Thank you for the suggestion. We agree that including modelling would allow assessing the results of this study in a more regional setting. However, it is out of scope for this study. We strongly agree that this is a very relevant comment, and there is a current, on-going project where this will be accounted for on a European scale. In addition, we are currently working on a study where the measurements done at Utö are compared to the MetCoOp model Harmonie-AROME.

1. Parts of the introduction seem to make a false dichotomy between VHF and AIS (e.g. 38-40, 52-53). Consider rephrasing.

Thank you for pointing this out. We understand the issue the referee is describing. We removed the mention of AIS frequency for 38-40 and added a specification on the line 52-53:

"..., other systems using the VHF frequency..."

Line-by-line comments as follows:

14-15: Provide some statistical metric to support the claim that "anomalous AIS observations were also found to coincide with the stronger and higher observed ducts"

Thank you for the comment. We have edited the sentence as follows:

"Anomalous AIS observations were also associated with stronger and higher ducts; when the duct height was 59 m, the occurrence rates were 90% and 95% for the 7 m and 30 m antenna, respectively."

41: Specify 'at distances of less than 1000 km.'

We thank the reviewer for pointing this out. It is an important specification to make. We have added it at the end of the sentence:

"However, at the AIS frequency 162 MHz, troposcatter and ducting are the most relevant factors resulting in anomalous signal propagation at distances of less than 1000 km."

53: Given the separate categorization of (1) AIS and (2) VHF, Chartier et al. (2022) belongs in the first group rather than the second.

Thank you for pointing this out; it has now been fixed.

223 (and elsewhere): Consider using a different term than 'horizon'. The manuscript makes sense if 'horizon' is interpreted as 'horizon of observability', but the most natural interpretation is 'the line at which the earth's surface and the sky appear to meet.'

Thank you for the suggestion. The natural interpretation does not account for the refraction by the atmosphere, while the horizon defined in the preprint is the horizon of observability under standard atmospheric conditions. We have defined the term horizon prior to using it (see L219 in the preprint).

---

## Author Comment (AC2)

**Response to Referee #2.**

Rautiainen, L., Johansson, M., Lensu, M., Tyynelä, J., Jalkanen, J.-P., Stenbäck, K., Lonka, H., and Laakso, L.: Studying anomalous propagation over marine areas using an experimental AIS receiver set-up, EGUsphere [preprint], <a href="https://doi.org/10.5194/egusphere-2025-1790">https://doi.org/10.5194/egusphere-2025-1790</a>, 2025.

We would like to thank Referee #2 for their thorough review and insightful comments. We have provided point-by-point responses to the comments below in red font.

**RC2 and RC3 - Anonymous Referee #2 - 10 Sept 2025**

The general idea and concept of that paper is certainly valid observing AIS signals at different heights and compare the occurrence, covered distances and directions to locally measured atmospheric parameters like temperature and humidity.

The location and general setup of the measurements is definitely suitable to investigate tropospheric propagations occurring in the VHF range.

The manuscript needs to be revised and refined, e.g., by sorting the content, moving rather supplementary materials from Results to the Appendix, and, in particular, by condensing the Methods and Results sections.

The current version of the paper feels rather lengthy, somewhat overloaded and appears to be rather an extended technical report than a full scientific paper for the subsequent reasons.

The actual discussion appears slim, it's rather a summary and an outlook, where in future such a setup could be used for.

Here it would be good to discuss the findings, e.g. local duct with ducts derived from weather model reanalysis / forecasts, which is likely out of the scope of this publication.

Much of the contents worth to place in the Discussion section are already covered in the Results section, where the pure description there is already quite detailed.

I miss a more extensive discussion and comparison to other publications in terms of duct altitudes, widths, covered ranges, seasonal occurrences etc.

The conclusions are short, but not too conclusive except the antenna mounted at higher altitudes covers a larger area for both "normal" and "anomalous" propagations and sea areas tend to have better chances to provide favorable conditions.

The use of some terms like "horizon" or "height" of the duct is not strictly clear and needs careful re-phrasing. For the latter, is this the center height of the duct?

We agree that the article is lengthy and appreciate the suggestions made by Referee #2 on how to fix this.

We also acknowledge that while including regional modelling would improve the study, it is out of scope of this observations-based study. Including regional modelling would only complicate and further lengthen the study. We currently have a study underway where the MetCoOp model Harmonie-Arome is studied alongside the Utö mast measurements, and another on-going project where the data is studied alongside the ERA5 reanalysis product which will hopefully address some of the concerns raised here and by Referee #1.

We see this study as a starting point for using AIS for ducting monitoring and establishing climatology. The current published studies have focused on the modelling of AIS signal (and VHF frequency in general) but less so on the climatology aspect. The discussion is slim because there is not much to compare our results to due to the novelty of the approach taken in this study. We previously published an article on 2 years of our mast and coastal radar measurements where a more detailed look and comparison was done (Rautiainen et al., 2025). Hence, there is no comparison of our duct characteristics with other studies in this study.

Additionally, the term "horizon" has been defined in L219 in the preprint while the term "duct height" has been defined in L182.

details:

L19 "it communicates" the system doesn't communicate, it is used for communication between...

Fixed as suggested

L20 "VHF signals propagate via LoS propgation" - re-phrase

Fixed

"VHF signals propagate via LoS"

L21 the separation troposphere and physical objects/surfaces needs re-phrasing..., e.g. tropospheric scattering is very common

**Fixed**

"...and interact with the troposphere and physical objects and surfaces via refraction, reflection, diffraction and scattering."

L33 when introducing M - refer to explanations in sect. 2.5

**Fixed.**

L36 What are "opportunistic effects"?

Where the increased visibility can result in tactical advantages. To simplify, we have edited the sentence:

"Due to both detrimental and opportunistic effects of ducting, ..."

**to:**

"Due to the potential impact on operational coverage, ..."

L38/39 This comment/statement for "VHF" is very general... you're mentioning here completely different processes, with much different occurrences and efficiencies (propagation loss), e.g. Earth-Moon-Earth is certainly for most applications irrelavant, especially for AIS, where a few W EIRP instead of several kW EIRP are required.

Otherwise you could also list here Field-Alligned-Irregularities, auroral backscatter, D- or E-region-iono-scatter, Transequatorial propagation etc etc etc...

Thank you for raising this concern. The sentence has been edited to:

"Besides ducting, there are other causes of anomalous propagation for the VHF channel."

L45 appears to general to me: I suspect radio sondes also fly over the sea, also airplanes collect measurements, though the flight heights might much above the typical tropospheril duct altitudes

Scarcity is used to note that while these exist, they do not cover the sea areas adequately in time or space. Radiosonde data is available but still very limited (typically every 6-12 hours) and the location of the measurements depends on the wind direction, while the

data collected by airplanes is not openly available. We agree that the statement is perhaps too general and edited the sentence as follows:

"Due to the scarcity of continuous measurements"

L62 "... behave in uexpected ways" means what?

The term "unexpected" is used to describe situations where the data does not follow the guidelines expected by the user, e.g. how often the messages are transmitted should correspond to the speed of the vessel, which is not always the case.

L71 "amsl" - I'd tend to write out the abbreviation once

Fixed.

L80 "GT" as above L71

Fixed.

L88-95 is this paragraph needed for the paper?

This paragraph was included to introduce some of the concepts that are referred to in the text when explaining which messages were used in which analysis, and so on. We appreciate the suggestion on how to condense the article and have removed the paragraph.

L107 "All the antenna cables were set at the same length of 120m." Do you really mean between the antenna and the receiver?

as an example, a 1/2" coaxial cable would have about 4dB loss, and it's a passive 2dBi antenna...

so the sensitivity of your setup is already limited by the long coaxial cable.

Yes. The cable length is 120 m as originally, we had three set-ups at different heights, and the cable length was decided to be kept the same for all three to avoid any differences caused by the cable losses. The cable used is LL400 and the cable losses calculated by the manufacturer for the 120 m cable is max. 17 dB (for 1 GHz). At the 160 MHz frequency the losses are likely to be less, about 6 dB/100 m.

We have edited the sentence as follows:

"Both antenna cables were set at the same length of 120 m, which introduces approximately 7 dB loss. Although this reduces the sensitivity, it ensures that the set-ups are comparable."

Fig1 Caption: add The VHF antennae" > are located/mounted < " at 30m and..."

Fixed.

L125 ORBCOMM global AIS data... I'm not too convinced you really needed that data for the study (Sect. 3.6)

what is the outcome of this comparison -> "what have we learnt"?

It feels like you wanted to >report

Figure 1. Same as Fig 12 but the colorbar starts at 5%.

For this study, we consider the method sufficient. For any future use, the method could be improved by decreasing the grid size, e.g. to nautical grid, and by involving SAR measurements as a further validation method. That would be an interesting study by itself.

L190-191 I can't quite follow. In Fig. 4 I see for the winter: increase of "rec.", basically constant "virtual aid" and reduced "preprocessed"

Thank you for pointing this out; it is indeed unclear. Fig 4 shows that towards the spring, the number of received messages increases but when removing only virtual aid messages, the number of received messages stays relatively constant.

**Edited the sentence to:**

"The number of discarded messages increases during winter and peaks in early spring, where nearly 50% is discarded during preprocess."

L193-194 "characteristics of the receiving antenna (e.g. height, sensitivity and power)..." why power???

Thank you for pointing this out; power should not be included here for the receiving antenna. This has been fixed.

L197 "...potential shadowing of antennae by other ships..." I'd guess this is negligible, if not a small sailing boat hides directly behind a large ship...?

It is likely negligible but theoretically possible. The atmospheric conditions being the most likely culprit.

L207- I'd suppress all the data below 10km (at least!) as the local harbour spoils the statistics and as you're rather after "anomalous" propagation these distances are not useful anyway.

The figure here is mainly included to illustrate the overall data and where from the messages are received prior to any exclusions. This justifies excluding the nearest 1 km from the analysis as the harbour on the island pollutes the data. Excluding 10 km from the data could potentially hide other forms and effects of anomalous propagation, such as subrefraction.

Fig7 and next are made for the entire dataset, not specific months, right? It's not clearly stated...

an additional x-axis in units of km would be helpful, log(distance) is not that easy to interprete for most

The 95th percentile analysis is done for the whole dataset. The following analyses based on the 95th percentile are also for the year.

To address this, we have added "... based on a year of data" to the Fig. 7 and Fig. 8 captions.

Fig9 is a good candidate for the appendix

Thank you for the suggestion on how to condense the study. We have moved Fig. 9 to the appendix.

Fig10 would this Fig or the content need a normalizing by the number of vessels available in the area ???

thinking of no. of vessels over the year and over the day

The number of vessels within 5 km from the station 2006-2020 is shown in Maragkidou et al. 2025. The number of vessels would increase with the increase in the area visible to the antenna, too.

Maragkidou, A., Grönholm, T., Rautiainen, L., Nikmo, J., Jalkanen, J.-P., Mäkelä, T., Anttila, T., Laakso, L., and Kukkonen, J.: Measurement report: The effects of SECA regulations on the atmospheric SO2 concentrations in the Baltic Sea, based on long-term observations on the Finnish island, Utö, Atmos. Chem. Phys., 25, 2443–2457, https://doi.org/10.5194/acp-25-2443-2025, 2025.

L275 occurrence daytime/nighttime - ok, so why is this? worth to e.g. discuss in the Discussion section

-> chances to form suitable temp. and humidity profiles / sea temp. is much more stable than ground/soil/rock, plus air radiative cooling etc.

so, near the archipelago area has less chances

Added a paragraph to the end of Discussion section:

"Similarly to findings by Norin (2022) and Rautiainen et al (2025), the AIS OH observations have diurnal and seasonal cycles. The diurnal cycle is found to result from the archipelago sector where the occurrence increases 35% from daytime to evening and night. This reflects the radiative cooling over land that creates a stable stratification over land at night, allowing for the ducts to form more readily at night. The sea surface temperature is more stable overnight and prevents the development of a similar diurnal cycle."

Fig11 another candidate for the appendix ... just the numbers in the text or a small table would be fine

Thank you for the suggestion on how to condense the study. Instead of showing daily percentiles, we have calculated the percentiles over the whole dataset and included them in a table (Table 1 in the new revised manuscript).

Fig12 the colobar is not too suitable, hard to see distinguish anything between 103 and 105 - either use a different colorbar, or log scale.

Thank you for pointing this out. We have changed to colorbar and it is now easier to separate the colours between the increments.

Sect 3.6 I'm not too convinced, what has been learnt from the global AIS, this probably needs to be clarified more clearly.

In that light, for the global AIS as it is comprised of spaceborne and terrestrial data -

during ducting, I'd bet quite some of the AIs signals are not received by the satellites, but at the same time more terrestrial detections will exist, which compensates?...

Thank you for the comment, please see the answer to L125 comment.

L305-306 I don't agree with this statement... it's actually contradicted with the following sentence.

Furthermore I think it's valid to assume the receiver will not only receive messages from local boats, but also long distance vessels sailing along the major routes.

I can't see why this shouldn't be the case in October.

Thank you for the comment. We have added the following clarifications:

"The extent of visibility increases with height and both antennae achieve great visibility along the busy ship routes. The visibility in September is much greater than in October. To address if the regions of low visibility are due to there simply being no vessels to receive messages from, or if the regions are out of range for the AIS antennae, the ORBCOMM global AIS data was also gridded..."

Fig15 another candidate for the appendix or to be removed as it's basically a scaled version of Fig14. What do we learn from this Fig? Again, just less observations/coverage. in the right panel, why are there distances near 0km, when it's about OH grids?

Thank you for the suggestion on how to condense the study. We have moved Fig. 15 to the appendix.

L327 "percentile is more sensitive to..." -> percentile of distances is more sensitive to...

Fixed.

L328 true, but this would mean it's a duct in specific directions and areas

Good point!

Edited the sentence to:

"e.g. a small number of ships during an hour could cause a spike in the hourly 95th percentile of distance, *indicating a duct in a specific area*."

L330 "However, it is unclear if the anomalous IAS" Why unclear, what else should it be?

Thus far in the study, it has not been shown to be caused by ducting, as the OH observations could also result from other processes, e.g. troposcatter.

L332 This valid test bears the assumption, that the receiver (Utö) is within the duct, but the duct could just be near and the receiver has a favorable incident angle to the duct. -> coming up later again and should be phrased and discussed

-> corresponding speculation in L351, L400

We have added the following sentences to follow L332 (in the preprint):

"It is important to note that the maximum measurement height of 59 m amsl limits the detection of elevated ducts. In addition, the mast is representative of local conditions, while ducts that could influence the signal propagation can exist within the horizon of the antennae, undetected by the mast measurements."

Fig16 somewhat hard to see details, especially in the 3rd panel

Thank you for pointing this out; we have made the figure wider. Hopefully this helps the interpretation of the figure!

Fig17 again, why are there distances near 0km? It's certainly not the type of ducting (over hundreds of km) you're aiming for

I'd also move the 2nd panel to the appendix, if needed.

Only the nearest 1 km is excluded from the study. The data is shown in 10 km intervals and hence there is 0-10 km interval where data is 1-10 km. We think that the  $2^{nd}$  panel should be included as it demonstrates the differences between heights.

Fig18 where is a reasonable discussion of that Fig? There are just statements here and later on...

the lower antenna will likely have much more obstacles than the higher antenna and therefore can't reach that easily low elevation ducts

Thank you for pointing this out. We have added the following sentences (bolded) to help the interpretation of the figure and to also include discussion from L343 (in preprint):

"Particularly, the highest 95th percentile distances seem to co-occur with the stronger and higher observed ducts. **To examine this more closely, the occurrence of AIS OH observations were studied against the duct height and duct strength (Fig. 15).** When the duct height is observed at 59 m, the 95th percentile of distance is increased 90% of the time for the 7 m antenna and 95% of the time for the 30 m antenna **while the share of**

stronger ducts also increases with the height of the duct. It appears that when the observed duct height is 32 or 59 m, AIS OH observations can be expected. Furthermore, the 7 m antenna observes less ducts than the 30 m antenna at all heights. However, the greatest difference occurs when the duct height is small. There are more obstacles to the 7 m antenna which can cause there to be less OH observations for the 7 m antenna when the duct height is low."

L364 "... is sensitive to number of ..." -> "... is sensitive to the number of ..."

**Fixed.**

L381 "These atmospheric duct parameters often include duct height, strength, thickness, and slope."

Yes, can you summarise your results, e.g. in table, preferred heights, thicknes etc.?

Thank you for the suggestion. We have previously published a more detailed analysis based on 2 years of the mast data (see Rautiainen et al. 2025). Due to the mast being limited to 59 m in height, we do not attempt to characterize ducts that would lead to AIS ducting in this study. The dataset would need to be supplemented with weather soundings.

Rautiainen, L., M. Lensu, V. Vakkari, J. Tyynelä, H. Kanarik, and L. Laakso, 2025: Marine Atmospheric Ducting Statistics Based on 2 Years of Coastal Surveillance Radar Observations. J. Appl. Meteor. Climatol., **64**, 63–76, https://doi.org/10.1175/JAMC-D-24-0096.1.

L383-384 I'd agree, can you perhaps already specify the contribution of this paper?

This study does not attempt at improving duct inversion studies but to assess how frequent AIS ducting is in the northern Baltic Sea, i.e. highlighting how much potential AIS has for monitoring ducting.

L386 "... that defines the signal ... defines -> describes ???

**Fixed**

L390-392 again, it's rather a summary, not a discussion... could you "speculate" why the occurrences are different?

different altitudes, thickness, gradients -> intensity etc...

This point was also raised by Referee #3 and as such we have written the same answer for both comments:

Our study cannot confidently state that the increased horizon (i.e. anomalous propagation) is only due to ducting, especially as our observations are limited to 59 m above the mean sea level. Based on the maximum wavelength (e.g. Kerr, 1951; Turton et al., 1988), X- and C-band generally require shallower (> 10 m) and weaker ducts than the VHF-band (> 100 m). While the X- and C-band are affected by evaporation and surface ducts, the VHF-band is more affected by elevated ducts, and our current set-up does not allow for detection of elevated ducts above 59 m.

We are planning a measurement campaign where soundings will be carried out at the measurement site which hopefully will shed some light on the issue. Meanwhile, there are no comparable studies that would allow us to compare our results. We hope that this study will inspire others to reproduce the study as the set-up is low-cost and simple to implement.

We have added the following clarification:

"The X- and C-band are more affected by the surface ducts (~10s of meters), while the VHF-band is affected by the elevated ducts (~100s of meters). Unfortunately, the set-up at Utö is limited to the height of 59 m, which omits the assessment of elevated ducts. For future analyses, including weather soundings or drone measurements to account for the heights above 60 m is needed."

Kerr, D. E., 1951: Propagation of Short Radio Waves. McGraw-Hill, 728 pp.

Turton, J. D., D. A. Bennetts, and S. F. G. Farmer, 1988: An introduction to radio ducting. *Meteor. Mag.*, **117**, 245–254.

L404 "... the occurrence of ducts underestimated ..." -> "... the occurrence of locally observed ducts underestimated ..."

Fixed

Sorry to be that blunt, I'm still struggeling to see what has been learnt from the 2 antenna setup...

Besides that, topics like the likelihood of ducts, existence of high-pressure systems (contant pressure path), suitable temperature / humidity profiles over various terrain and land/sea should be discussed also in the perspective to other publications.

We agree that this is a very important aspect but unfortunately out of scope for this study. We compare our observed ducts to other publications in Rautiainen et al., 2025, and we have an ongoing project that will investigate this in more detail.

Rautiainen, L., M. Lensu, V. Vakkari, J. Tyynelä, H. Kanarik, and L. Laakso, 2025: Marine Atmospheric Ducting Statistics Based on 2 Years of Coastal Surveillance Radar Observations. *J. Appl. Meteor. Climatol.*, **64**, 63–76, https://doi.org/10.1175/JAMC-D-24-0096.1.

The signficance and importance of the work and data needs to be stressed in comparison to other publications.

We agree that the significance and importance of the work could be more clearly stated in the paper. They arise from it being the only long-term study on AIS signal ducting, and there is a limited amount of ducting research published from the Baltic Sea region in general. Often studies focus on e.g. communication links between two places, in this case a large amount of data is collected and analysed. Lastly, the study also includes a comparison with the global AIS dataset which has not been done in this context before.

Again, I certaily support publishing the observations, but the manuscript needs to be revised significantly, condensed and strengthening of the Discussion and Conclusions.

We thank Referee #2 for their time and the insightful comments that truly helped us condense and improve the manuscript!

Further publications perhaps worth to look up:

Salamon, S.J., Hansen, H.J. and Abbott, D. (2015), Modelling radio refractive index in the atmospheric surface layer. Electron. Lett., 51: 1119-1121. https://doi.org/10.1049/el.2015.0195

P. VALTR, P. PECHAČ, TROPOSPHERIC REFRACTION MODELING USING RAY-TRACING AND PARABOLIC EQUATION

https://www.radioeng.cz/fulltexts/2005/05\_04\_098\_104.pdf

Ao, C. O. (2007), Effect of ducting on radio occultation measurements: An assessment based on high-resolution radiosonde soundings, Radio Sci., 42, RS2008, doi:10.1029/2006RS003485.

Tang et al., Atmospheric Ducts Inversion with Over-the-Horizon Propagation of Automatic Identification System Signals

APSIPA Transactions on Signal and Information Processing, 2024, 13, e11

https://www.nowpublishers.com/article/OpenAccessDownload/SIP-20230078

Wolinsky-Mancini et al., 4th URSI AT-RASC, Gran Canaria, 19 – 24 May 2024

https://www.ursi.org/proceedings/procAT24/papers/0337.pdf

**RC3 – Anonymous Referee #2**

Fig12 I suggest to either swap the upper and lower panels, or at least to mark the months Sept./Oct., alternatively keep the order and mark "quiet" and "anomalous/enhanced" propagation

Thank you for the suggestion. We swapped the panels as suggested.

**another reference:**

M. Banafaa and A. H. Muqaibel, "Tropospheric Ducting: A Comprehensive Review and Machine Learning-Based Classification Advancements," in IEEE Access, vol. 13, pp. 22510-22534, 2025, doi: 10.1109/ACCESS.2025.3537160.

---

## Author Comment (AC3)

**Response to Referee #3.**

Rautiainen, L., Johansson, M., Lensu, M., Tyynelä, J., Jalkanen, J.-P., Stenbäck, K., Lonka, H., and Laakso, L.: Studying anomalous propagation over marine areas using an experimental AIS receiver set-up, EGUsphere [preprint], <a href="https://doi.org/10.5194/egusphere-2025-1790">https://doi.org/10.5194/egusphere-2025-1790</a>, 2025.

We thank Referee #3 for their time and the insightful comments that improved this manuscript. We have provided our responses to each comment below indicated with red font.

**RC4 – Anonymous Referee #3 - 19 Sept 2025**

An interesting manuscript showing ideas and possibilities to monitor atmospheric ducting conditions (near) real time, using relatively low-cost, off-the-shelf hardware.

The manuscript feels a bit long with a lot of details. The authors should consider leaving out some details in following chapters:

We thank the Referee for the valuable suggestions on how to condense the study. The length of the study was also pointed out by the other Referees. We have done the changes accordingly:

- remove/shortening last paragraph of chapter 2.1,
  Done
- shorten chapter 2.2, ex details on sensors later removed are not necessary, L106-110

Done

• Remove chapter 2.5 and introduce eq 4 and 5 in introduction, second paragraph. We removed the equations from chapter 2.5 and instead cited our previous studies where the same calculations were done. This significantly shortened the chapter.

The authors should be consequent when describing the 7 m and 30 m receivers. In some figures 30 is described first, while sometimes the 7 m receiver is described/plotted first. In figs 3, 7, 9, 12, 13, 15 the 7 m receiver is mentioned first, while in figs. 4, 5, 6, 10, 11, 16, 17, 18 the 30 m receiver is described/plotted first. This might be confusing to the reader. Suggestion is to consequently describe/plot 7 m first and 30 m second.

We thank the Referee for the suggestion; it is indeed quite inconsistent. We have fixed these as instructed.

In Discussion line 390-393 comparing your findings with Rautiainen 2025 and Norin 2023 where AIS is more often affected by ducting than X and C band radars: This difference needs a comment or discussion. Usually, higher frequencies (X and C band) are more affected by ducting and require lower/shallower atmospheric ducts than lower frequencies (VHF) to be captured in the duct.

This point was also raised by Referee #2 and as such we have written the same answer for both comments:

Our study cannot confidently state that the increased horizon (i.e. anomalous propagation) is only due to ducting, especially as our observations are limited to 59 m above the mean sea level. Based on the maximum wavelength (e.g. Kerr, 1951; Turton et al., 1988), X- and C-band generally require shallower (> 10 m) and weaker ducts than the VHF-band (> 100 m). While the X- and C-band are affected by evaporation and surface ducts, the VHF-band is more affected by elevated ducts, and our current set-up does not allow for detection of elevated ducts above 60 m.

We are planning a measurement campaign where soundings will be carried out at the measurement site which hopefully will shed some light on the issue. Meanwhile, there are no comparable studies that would allow us to compare our results. We hope that this study will inspire others to reproduce the study as the set-up is low-cost and simple to implement.

**We have added the following:**

"The X- and C-band are more affected by the surface ducts (~10s of meters), while the VHF-band is affected by the elevated ducts (~100s of meters). Unfortunately, the set-up at Utö is limited to the height of 59 m, which omits the assessment of elevated ducts. For future analyses, including weather soundings to account for the heights above 60 m is needed."

Kerr, D. E., 1951: Propagation of Short Radio Waves. McGraw-Hill, 728 pp.

Turton, J. D., D. A. Bennetts, and S. F. G. Farmer, 1988: An introduction to radio ducting. *Meteor. Mag.*, **117**, 245–254.

**Details, line by line:**

53: Consider using Gunashekar et al. 2010, 'Long-term statistics related to evaporation duct propagation of 2 GHz radio waves in the English Channel' instead of Gunashekar et al 2006.

We appreciate the very interesting article recommended by the reviewer. We added a line to also acknowledge UHF frequency and included the recommended article. This article will help us in our future studies also, thank you!

75: Clarify that M-profiles are derived from the mast measurements of T, RH.

**Fixed.**

Fig. 1 a) Arrows pointing at the AIS receivers.

**Done.**

144: Is the maximum estimated AIS range for the 7 m receiver correct? Text says 65-80 km, while in Fig 3 it seems to be approx. 50-75 km. Or do I interpret Fig 3 wrong? I interpret the maximum AIS rang to be where the dashed and dotted green and pink lines cross the black -115 dB line. For 30 m receiver text and Fig 3 are coincidence.

We thank the reviewer for noticing this error, it should say 50-75 km. We have fixed this in the text.

171: Remove the part "the refractivity N". You have already introduced 'M' and said that "For all practical purposes, M is used …" in line 168. The sentence could then be 'In order to study if ducting influences the AIS range observed in Utö, the modified refractivity M (Eq. 4 and 5) were calculated ……'

Fixed.